# Identifiability of total effects from abstractions of time series causal graphs

**Charles K. Assaad**[1,2]    **Emilie Devijver**[3]    **Eric Gaussier**[3]    **Gregor Gössler**[4]    **Anouar Meynaoui**[5]

[1]Sorbonne Université, INSERM, Institut Pierre Louis d'Epidémiologie et de Santé Publique, F75012, Paris, France
[2]EasyVista, F38000, Grenoble, France
[3]Univ Grenoble Alpes, CNRS, Grenoble INP, LIG, F38000, Grenoble, France
[4]Univ. Grenoble Alpes, INRIA, CNRS, Grenoble INP, LIG, F38000, Grenoble, France
[5]Université of Rennes 2, F35000, Rennes, France

## Abstract

We study the problem of identifiability of the total effect of an intervention from observational time series in the situation, common in practice, where one only has access to abstractions of the true causal graph. We consider here two abstractions: the extended summary causal graph, which conflates all lagged causal relations but distinguishes between lagged and instantaneous relations, and the summary causal graph which does not give any indication about the lag between causal relations. We show that the total effect is always identifiable in extended summary causal graphs and provide sufficient conditions for identifiability in summary causal graphs. We furthermore provide adjustment sets allowing to estimate the total effect whenever it is identifiable.

## 1 INTRODUCTION

Over the last century and across numerous disciplines, experimentation has emerged as a potent methodology for estimating without bias the total effect of an intervention on a specific component of a given system [Neyman et al., 1990]. However, experimentation can be costly, unethical or even unfeasible. Both researchers and experts are thus interested in estimating the effect of an intervention directly from observational data. This can be done under some assumptions when relying on a complete causal graph [Pearl et al., 2000], and typically relies on two sequential steps: identifiability and estimation [Pearl, 2019]. The identifiability step involves distinguishing cases where a solution is possible and, when it exists, providing an estimand - an expression enabling the estimation of intervention effects from observational data. The subsequent step involves the actual estimation of this estimand from the available data.

The identifiability step received much attention for non-temporal causal graphs [Pearl, 1993, 1995, Spirtes et al., 2000, Pearl et al., 2000, Shpitser and Pearl, 2008]. For abstraction of causal graphs, Perkovic [2020] derived necessary and sufficient conditions for identifying total effects in maximally oriented partially directed acyclic graphs and Anand et al. [2023] provided necessary and sufficient conditions when dealing with a directed acyclic graphs, where each vertex represent a cluster of variables and where relationships between clusters of variables are specified, but relationships between the variables within a cluster are left unspecified.

For temporal causal graph, Blondel et al. [2016] developed the do-calculus for the full-time causal graphs (FTCGs, Figure 1.1a). However, in dynamic systems, experts have difficulties in building full time causal graphs, while they can usually build an abstraction of those graphs such as an extended summary causal graph (ESCG, as in Figure 1.1b) where all lagged causal relations are conflated but lagged and instantaneous relations are clearly distinguished or such as a summary causal graph (SCG, as in Figure 1.1c) where all temporal information is omitted. Assuming no instantaneous relations, Eichler and Didelez [2007] demonstrated that the total effect is identifiable from an ESCG or an SCG, and Assaad et al. [2023] established identifiability in the presence of instantaneous relations for acyclic SCGs. Ferreira and Assaad [2024] addressed the identifiability problem for general SCGs, including cycles and instantaneous relations for the direct effect; however, the identifiability of total effects in this context remains unexplored.

Our main contributions consist in demonstrating, under causal sufficiency, that the total effect is always identifiable when working with an extended summary causal graph and providing sufficient conditions for identifying the total effect when working with a summary causal graph. The main difficulty lies in the fact that these abstractions may represent different full-time causal graphs with potentially different skeletons and orientations.

The remainder of the paper is structured as follows: Section

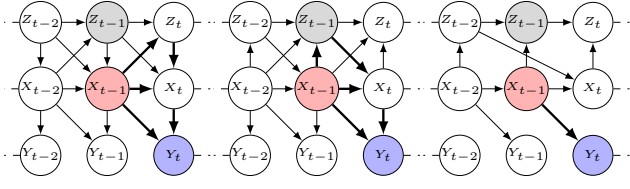

(a) Three FTCGs, $\mathcal{G}_1^f$, $\mathcal{G}_2^f$ and $\mathcal{G}_3^f$.

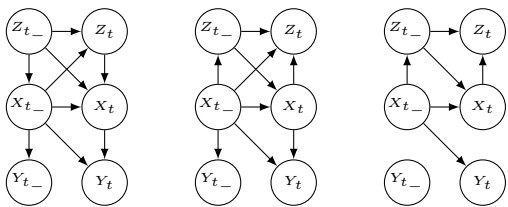

(b) Three ESCGs, $\mathcal{G}_1^e$, $\mathcal{G}_2^e$ and $\mathcal{G}_3^e$, resp. derived from $\mathcal{G}_1^f$, $\mathcal{G}_2^f$ and $\mathcal{G}_3^f$.

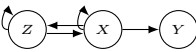

(c) The SCG $\mathcal{G}^s$, derived from any FTCG in (a) and any ESCG in (b).

Figure 1.1: Illustration: (a) three FTCGs, (b) three ESCGs derived from them, (c) the SCG which can be derived from any FTCG in (a) and any ESCG in (b). Consider $f(y_t|do(x_{t-1}))$, red vertex: the variable we intervene on, blue vertex: the response we are considering. Bold edges correspond to directed paths from $X_{t-1}$ to $Y_t$, and gray vertices correspond to nodes with different status depending on the FTCG (see Definition 6).

2 introduces the main notions, Section 3 presents the problem setup, identifiability conditions in ESCGs and SCGs are respectively presented in Sections 4 and 5. Section 6 discusses real applications for our theoretical results, and Section 7 concludes the paper. Omitted proofs can be found in the Supplementary Material.

## 2   PRELIMINARIES

**Graph notions**   For a *directed acyclic graph* $\mathcal{G}$, a *path* from $X$ to $Y$ in $\mathcal{G}$ is a sequence of distinct vertices $<X,\ldots,Y>$ in which every pair of successive vertices is adjacent. A *directed path* from $X$ to $Y$ is a path from $X$ to $Y$ in which all edges are directed towards $Y$ in $\mathcal{G}$, that is $X \to \ldots \to Y$. A *backdoor path* between $X$ and $Y$ is a path between $X$ and $Y$ with an arrowhead into $X$ in $\mathcal{G}$. If $X \to Y$, then $X$ is a *parent* of $Y$. If there is a directed path from $X$ to $Y$, then $X$ is an *ancestor* of $Y$, and $Y$ is a *descendant* of $X$. A vertex counts as its own descendant and as its own ancestor. The sets of parents, ancestors and descendants of $X$ in $\mathcal{G}$ are denoted by $\mathrm{Par}(X,\mathcal{G})$, $\mathrm{Anc}(X,\mathcal{G})$ and $\mathrm{Desc}(X,\mathcal{G})$ respectively. If a path $\pi$ contains $X \to Z \leftarrow Y$ as a subpath, then $Z$ is a *collider* on $\pi$. A vertex $Z$ is a

*definite non-collider* on a path $\pi$ if the edge $X \leftarrow Z$, or the edge $Z \to Y$ is on $\pi$. A vertex is of *definite status* on a path if it is a collider, a definite non-collider or an endpoint on the path. A path $\pi$ is of *definite status* if every vertex on $\pi$ is of definite status. A path $\pi$ from $X$ to $Y$ of definite status is *active* given a vertex set $\mathcal{Z}$, with $X, Y \notin \mathcal{Z}$ if every definite non-collider on $\pi$ is not in $\mathcal{Z}$, and every collider on $\pi$ has a descendant in $\mathcal{Z}$. Otherwise, $\mathcal{Z}$ *blocks* $\pi$. By a slight abuse of notation, we denote $\mathcal{G}\backslash\{Y\}$ as the subgraph of $\mathcal{G}$ when removing the vertex $Y$ and its corresponding edges. Lastly, the *skeleton* of a graph corresponds to all vertices and edges of the graph without considering edge orientations.

For a *directed graph* $\mathcal{G}$, a directed path from $X$ to $Y$ and a directed path from $Y$ to $X$ form a *directed cycle* in $\mathcal{G}$. A self-loop on $X$ also forms a directed cycle. We denote by $Cycles(X,\mathcal{G})$ the set of all directed cycles containing $X$ in $\mathcal{G}$, and by $Cycles^>(X,\mathcal{G})$ the subset of $Cycles(X,\mathcal{G})$ with at least two different vertices (i.e., excluding self-loops). In addition, all notions introduced before for directed acyclic graphs hold for *directed graphs*, with potential cycles. To avoid any ambiguity we would like to make some clarifications. For a *directed graph* $\mathcal{G}$, a *backdoor path* between $X$ and $Y$ is a path between $X$ and $Y$ which starts by either $X \leftarrow$ or $X \leftrightarrows$. A path is blocked by an empty set if there exists a vertex $Z$ such that $\to Z \leftarrow$ is on the path. Note that the above does not hold for $\leftrightarrows Z \leftarrow$, $\to Z \leftrightarrows$ or $\leftrightarrows Z \leftrightarrows$. For clarity, whenever a path is blocked by an empty set in a directed graph we will say that it is $\sigma$-blocked[1]. Note that $X \rightleftarrows Y$ and $X \leftarrow Y$ are the only $\sigma$-active backdoor paths of size 2 in $\mathcal{G}$.

If each vertex in a directed acyclic graph corresponds to an observed variable then, given an ordered pair of vertices $(X,Y)$ in $\mathcal{G}$, a set of vertices $\mathcal{Z}$ satisfies the *standard backdoor criterion* relative to $(X,Y)$ if no vertex in $\mathcal{Z}$ is a descendant of $X$, and $\mathcal{Z}$ blocks every backdoor path between $X$ and $Y$.

**Causal graphs in time series**   Consider $\mathcal{V}$ a set of $p$ observational time series and $\mathcal{V}^f = \{\mathcal{V}_{t-\ell}|\ell \in \mathbb{Z}\}$ the set of temporal instances of $\mathcal{V}$ where $\mathcal{V}_{t-\ell}$ correspond to the variables of the time series at time $t-\ell$. We suppose that the time series are generated from an *unknown* dynamic structural causal model (DSCM, Rubenstein et al. [2018]), an extension of structural causal models (SCM, Pearl et al. [2000]) to time series. This DSCM defines a full-time causal graph (FTCG, see below) which we call the *true* FTCG [Runge et al., 2019, Runge, 2021, Assaad et al., 2022a] and a joint distribution $P$ over its vertices which we call the *true* probability distribution.

---

[1]The notion of $\sigma$-blocked path by a set $\mathcal{Z}$ is a generalization of the notion of blocked path by a set $\mathcal{Z}$ (which was introduced for directed acyclic graphs) to directed graphs [Forré and Mooij, 2017]. These two notions becomes equivalent when $\mathcal{Z} = \emptyset$. In this paper, we will use the notion of $\sigma$-blocked only when $\mathcal{Z} = \emptyset$.

**Definition 1** (Full-time causal graph (FTCG), Figure 1.1a).
*Let $\mathcal{V}$ be a set of $p$ observational time series and $\mathcal{V}^f = \{\mathcal{V}_{t-\ell} | \ell \in \mathbb{Z}\}$. The full-time causal graph (FTCG) $\mathcal{G}^f = (\mathcal{V}^f, \mathcal{E}^f)$ representing a given DSCM is defined by: $X_{t-\gamma} \to Y_t \in \mathcal{E}^f$ if and only if $X$ directly causes $Y$ at time $t$ with a time lag of $\gamma > 0$ if $X = Y$ and with a time lag of $\gamma \geq 0$ for $X \neq Y$.*

As common in causality studies on time series, we consider in the remainder acyclic FTCGs with potential instantaneous causal relations. Note that acyclicity is guaranteed for relations between variables at different time stamps. In addition, note that for any time series $X$, $\forall i > 0$, $X_{t-i}$ can cause $X_t$; for example, the stock price yesterday can affect the stock price today. We furthermore assume causal sufficiency:

**Assumption 1** (Causal sufficiency). *There is no hidden common cause between any two observed variables.*

In practice, it is usually impossible to work with FTCGs and people have resorted to simpler causal graphs, exploiting the fact that causal relations between time series hold throughout time, as formalized in the following assumption which allows one to focus on a finite number of past slices, given by the maximum lag. We fix it to $\gamma_{\max}$ in the remainder.

**Assumption 2** (Consistency throughout time). *All the causal relationships in the the FTCG $\mathcal{G}^f$ remain constant in direction and magnitude throughout time[2].*

Experts are used to working with abstractions of causal graphs which summarize the information into a smaller graph that is interpretable, often with the omission of precise temporal information. We consider in this study two known causal abstractions for time series, namely *extended summary causal graphs* and *summary causal graphs*. An extended summary causal graph [Assaad et al., 2022c] distinguishes between past time slices, denoted as $\mathcal{V}^e_{t-}$, and present time slices, denoted as $\mathcal{V}^e_t$, thus enabling the differentiation between lagged and instantaneous causal relations.

**Definition 2** (Extended summary causal graph (ESCG), Figure 1.1b). *Let $\mathcal{G}^f = (\mathcal{V}^f, \mathcal{E}^f)$ be an FTCG built from the set of time series $\mathcal{V}$ satisfying Assumption 2 with maximal temporal lag $\gamma_{\max}$. The extended summary causal graph (ESCG) $\mathcal{G}^e = (\mathcal{V}^e, \mathcal{E}^e)$ associated to $\mathcal{G}^f$ is given by $\mathcal{V}^e = (\mathcal{V}^e_{t-}, \mathcal{V}^e_t)$ and $\mathcal{E}^e$ defined as follows:*

- *for any $X$ in $\mathcal{V}$, we define two vertices, $X_{t-}$ and $X_t$, respectively in $\mathcal{V}^e_{t-}$ and $\mathcal{V}^e_t$;*

---

[2]In our context we consider a dynamic system with several univariate observational time series, thus the problem of finding a unique total effect would be ill-posed if Assumption 2 is not satisfied since violating the assumption would mean that the total effect would change over time.

- *for all $X_t, Y_t \in \mathcal{V}^e_t$, $X_t \to Y_t \in \mathcal{E}^e$ if and only if $X_t \to Y_t \in \mathcal{E}^f$;*

- *for all $X, Y \in \mathcal{V}^e_{t-}$, $X_{t-} \to Y_t \in \mathcal{E}^e$ if and only if there exists at least one temporal lag $0 < \gamma \leq \gamma_{\max}$ such that $X_{t-\gamma} \to Y_t \in \mathcal{E}^f$.*

*In that case, we say that $\mathcal{G}^e$ is derived from $\mathcal{G}^f$.*

At a higher level of abstraction, a summary causal graph [Peters et al., 2013, Meng et al., 2020, Assaad et al., 2022a,b] represents causal relationships among time series, regardless of the time delay between the cause and its effect.

**Definition 3** (Summary causal graph (SCG), Figure 1.1c).
*Let $\mathcal{G}^f = (\mathcal{V}^f, \mathcal{E}^f)$ be an FTCG built from the set of time series $\mathcal{V}$ satisfying Assumption 2 with maximal temporal lag $\gamma_{\max}$. The summary causal graph (SCG) $\mathcal{G}^s = (\mathcal{V}^s, \mathcal{E}^s)$ associated to $\mathcal{G}^f$ is such that*

- *$\mathcal{V}^s$ corresponds to the set of time series $\mathcal{V}$,*

- *$X \to Y \in \mathcal{E}^s$ if and only if there exists at least one temporal lag $0 \leq \gamma \leq \gamma_{\max}$ such that $X_{t-\gamma} \to Y_t \in \mathcal{E}^f$.*

*In that case, we say that $\mathcal{G}^s$ is derived from $\mathcal{G}^f$ as well as from the ESCG derived from $\mathcal{G}^f$.*

Since an FTCG is assumed to be a directed acyclic graph, an ESCG is inherently a directed acyclic graph. In contrast, an SCG is a directed graph as it may include directed cycles and even self-loops. For example, the three FTCGs in Figure 1.1a and the three ESCGs in Figure 1.1b are acyclic, while the SCG in Figure 1.1c has a cycle. We use the notation $X \rightleftarrows Y$ to indicate situations where there are time lags where $X$ causes $Y$ and other lags where $Y$ causes $X$. Additionally, if an SCG is an abstraction of an ESCG, in cases where there is no instantaneous relation, ESCGs and SCGs convey the same information.

It is worth noting that if there is a single ESCG or SCG derived from a given FTCG, different FTCGs, with possibly different orientations and skeletons, can yield the same ESCG or SCG. For example, the SCG in Figure 1.1c can be derived from any FTCG and any ESCG in Figures 1.1a and 1.1b, even though they may have different skeletons (for example, $\mathcal{G}^f_1$ and $\mathcal{G}^f_3$ or $\mathcal{G}^e_1$ and $\mathcal{G}^e_3$) and different orientations (for example, $\mathcal{G}^f_1$ and $\mathcal{G}^f_2$ or $\mathcal{G}^e_1$ and $\mathcal{G}^e_2$). Therefore, even if each vertex in an FTCG is assumed to represent a single observed variable, a vertex in the past slice of an ESCG represent a set of variables while a vertex in the present time slice represents a single variable, and a vertex in the SCG corresponds to a time series. In the remainder, for a given ESCG or SCG $\mathcal{G}$, we call any FTCG from which $\mathcal{G}$ can be derived as a *candidate FTCG* for $\mathcal{G}$. For example, in Figure 1.1, $\mathcal{G}^f_1$, $\mathcal{G}^f_2$ and $\mathcal{G}^f_3$ are all candidate FTCGs for $\mathcal{G}^s$. The set of all candidate FTCGs for $\mathcal{G}$ is denoted by $\mathcal{C}(\mathcal{G})$.

# 3   PROBLEM SETUP

We focus in this paper on the *total effect* [Pearl et al., 2000] of the *singleton* variable $X_{t-\gamma}$ on the *singleton* variable $Y_t$, written $P(Y_t = y_t|do(X_{t-\gamma} = x_{t-\gamma}))$ (as well as $P(y_t|do(x_{t-\gamma}))$ by a slight abuse of notation), *when the only knowledge one has of the underlying DSCM consists in the ESCG or SCG derived from the unknown, true FTCG.* $Y_t$ corresponds to the response and $do(X_{t-\gamma} = x_{t-\gamma})$ represents an intervention (as defined in Pearl et al. [2000] and Eichler and Didelez [2007, Assumption 2.3]) on the variable $X$ at time $t - \gamma$, with $\gamma \geq 0$.

The above setting is very common in practice and entails that one neither knows the true FTCG nor the true probability distribution. Futhermore, even if one has access to observed data, in practice such observations are finite, which prevents one from discovering the true FTCG, and even from detecting it in the set of candidate FTCGs, as no existing causal discovery method is guaranteed to yield the true FTCG in the finite data setting [Aït-Bachir et al., 2023]. In the purely theoretical context of infinite data, discovering the true FTCG is only possible with additional assumptions, beyond the scope of this study [Assaad et al., 2022b].

Each candidate FTCG proposes a particular decomposition of the true joint probability distribution which is given by the standard recursive decomposition that characterizes Bayesian networks. Not all decompositions are however correct with respect to the true probability distribution $P$.

In general, a total effect $P(y_t \mid do(x_{t-\gamma}))$ is said to be identifiable from a graph if it can be uniquely computed with a do-free formula from the observed distribution [Pearl, 1995, Perkovic, 2020]. In our context, this means that the same do-free formula should hold in all candidate FTCG so as to guarantee that it holds for the true one.

**Definition 4** (Identifiability of total effects in ESCGs and SCGs). *In a given ESCG or SCG $\mathcal{G}$, $P(y_t \mid do(x_{t-\gamma}))$ is* identifiable *iff it can be rewritten with a do-free formula that is valid for any FTCG in $\mathcal{C}(\mathcal{G})$.*

One way to rewrite $P(y_t \mid do(x_{t-\gamma}))$ with a do free-formula is by finding an adjustment set of variables for which:

$$P(y_t|do(x_{t-\gamma})) = \sum_{\mathbf{z}} P(y_t|x_{t-\gamma}, \mathbf{z})P(\mathbf{z}). \qquad (1)$$

Whenever a set of variables satisfy Equation (1), we call it a *valid adjustment* set. The standard backdoor criterion, introduced in Pearl [1995], allows one to obtain valid adjustment sets using the true FTCG. We provide here another version of the backdoor criterion that allows us to find a valid adjustment set given all candidate FTCGs without knowing which one is the true FTCG.

**Definition 5** (Backdoor criterion over all candidate FTCGs). *Let $\mathcal{G} = (\mathcal{V}, \mathcal{E})$ be an ESCG or SCG. A set of vertices $\mathcal{Z}$*

satisfies *the* backdoor criterion over all candidate FTCGs *relative to $(X_{t-\gamma}, Y_t)$ if*

*(i)* $\mathcal{Z}$ *blocks all backdoor paths between $X_{t-\gamma}$ and $Y_t$ in any FTCG in $\mathcal{C}(\mathcal{G})$,*

*(ii)* $\mathcal{Z}$ *does not contain any descendant of $X_{t-\gamma}$ in any FTCG in $\mathcal{C}(\mathcal{G})$.*

Note that when there is no backdoor path between $X_{t-\gamma}$ and $Y_t$ in any FTCG in $\mathcal{C}(\mathcal{G})$, $\mathcal{Z} = \emptyset$ satisfies the backdoor criterion over all candidate FTCGs.

The backdoor criterion over all candidate FTCGs is sound for the identification of the total effect $P(y_t|do(x_{t-\gamma}))$ in an ESCG or SCG, as stated in the following corollary that can be deduced from [Pearl, 1995, Theorem 1].

**Corollary 1.** *Let $X$ and $Y$ be distinct vertices in an ESCG or SCG $\mathcal{G}$ of a DSCM with true (unknown) probability $P$. Under Assumptions 1 and 2 for $\mathcal{G}$, if there exists a set $\mathcal{Z}$ satisfying the backdoor criterion over all possible FTCGs relative to $(X_{t-\gamma}, Y_t)$, then the total effect of $X_{t-\gamma}$ on $Y_t$ is identifiable in $\mathcal{G}$, and $\mathcal{Z}$ is a valid adjustment set for the formulae given in Equation (1).*

However, enumerating all candidate FTCGs is computationally expensive [Robinson, 1977], even when considering the constraints given by an ESCG or an SCG.

Formally, we address the following technical problem:

**Problem 1.** *Consider an ESCG or an SCG $\mathcal{G}$ and the total effect $P(y_t|do(x_{t-\gamma}))$. We aim to find out conditions to identify $P(y_t|do(x_{t-\gamma}))$ when having access solely to an ESCG or an SCG without enumerating all candidate FTCGs in $\mathcal{C}(\mathcal{G})$.*

## Remarks

1. Our context is different from the one considered in Perkovic [2020] since the graphs we have to consider for a given ESCG or SCG, namely the candidate FTCGs, may have different skeletons and may not all be compatible with the true underlying distribution. Furthermore, in ESCGs and SCGs, each vertex does not necessarily correspond to a single variable.

2. Our context is different from the one considered in Anand et al. [2023]. They consider cluster of variables, even for the response and the intervention variable, while we are interested in the total effect $P(y_t \mid do(x_{t-\gamma}))$ where the response variable and the intervention variable are singletons. Furthermore, we may have cycles in the SCGs, while they assume acyclic graphs.

3. The cycles that we consider in this work, namely in SCGs, do not hold the same conceptual meaning as the cycles considered in Bongers et al. [2021], as in our

case, cyclicity comes from the abstraction of an acyclic graph.

# 4 IDENTIFIABILITY IN ESCG

The total effect is always identifiable by adjustment in ES-CGs, as stated in the following theorem.

**Theorem 1.** *(Identifiability in ESCG) Consider an ESCG $\mathcal{G}^e$. Under Assumptions 1 and 2 for $\mathcal{G}^e$, the total effect $P(y_t|do(x_{t-\gamma}))$ is identifiable in $\mathcal{G}^e$ for any $\gamma \geq 0$. Furthermore, the set*

$$\mathcal{B}_\gamma = \{(Z_{t-\gamma-\ell})_{1\leq\ell\leq\gamma_{\max}} | Z_{t-} \in Par(X_t, \mathcal{G}^e)\}$$
$$\cup \{Z_{t-\gamma} | Z_t \in Par(X_t, \mathcal{G}^e)\},$$

*is a valid adjustment set for $P(y_t|do(x_{t-\gamma}))$ for the formulae given in Equation (1).*

If $\mathcal{B}_\gamma$ is a valid adjustment set, it may still be very large. Additional adjustment sets, potentially smaller than $B_\gamma$, can however be obtained in the densest candidate FTCG, which is the candidate FTCG which contains all potential edges and is thus maximal in the number of edges.

**Proposition 1.** *Consider an ESCG $\mathcal{G}^e$ and a maximal lag $\gamma_{\max}$ and let $\gamma \geq 0$. Any adjustment set $\mathcal{B}'_\gamma$ for the total effect $P(y_t|do(x_{t-\gamma}))$ that satisfies the standard backdoor criterion on the densest candidate FTCG in $\mathcal{C}(\mathcal{G}^e)$ is a valid adjustment set for the total effect. In addition, $\mathcal{B}_\gamma$ is a valid adjustment set with respect to the standard backdoor criterion on the densest candidate FTCG.*

Note however that smaller (in the number of variables) adjustment sets may exist in the true FTCG when it is different from the densest candidate FTCG.

# 5 IDENTIFIABILITY IN SCG

In this section, we start by presenting the main result of the paper which provides sufficient conditions for identifying the total effect only by using an SCG and providing an adjustment set that can be used whenever the sufficient conditions are satisfied. Then we provide another adjustment set that is more suitable in practice. Finally, we discuss several examples where the total is not identifiable using an SCG.

Note that we are only considering sufficient conditions because the backdoor criterion is not complete, meaning it does not provide all possible valid adjustment sets. Therefore, the backdoor criterion over all candidate FTCGs is not necessarily complete.

## 5.1 MAIN RESULT: SUFFICIENT CONDITIONS FOR IDENTIFIABILITY

We provide sufficient conditions[3] for the identifiability in SCG. Recall that $Cycles(X, \mathcal{G}^s)$ is the set of all directed cycles containing $X$ in $\mathcal{G}^s$, and $Cycles^>(X, \mathcal{G}^s)$ is the subset where cycles contain at least 2 different vertices.

**Theorem 2.** *(Identifiability in SCG) Consider an SCG $\mathcal{G}^s = (\mathcal{V}^s, \mathcal{E}^s)$ associated with a DSCM with true (unknown) probability distribution $P$. Under Assumptions 1 and 2, the total effect $P(y_t|do(x_{t-\gamma}))$, with $\gamma \geq 0$, is identifiable if $X \notin Anc(Y, \mathcal{G}^s)$ or $X \in Anc(Y, \mathcal{G}^s)$ and none of the following holds:*

1. *$\gamma \neq 0$ and $Cycles^>(X, \mathcal{G}^s\backslash\{Y\}) \neq \emptyset$, or*

2. *there exists a $\sigma$-active backdoor path*

$$\pi^s = \langle V^1 = X, \cdots, V^n = Y \rangle$$

*from $X$ to $Y$ in $\mathcal{G}^s$ such that $\langle V^2, \cdots, V^{n-1} \rangle \subseteq Desc(X, \mathcal{G}^s)$ and one of the following holds:*

   (a) *$n > 2$, i.e. $\langle V^2, \cdots, V^{n-1} \rangle \neq \emptyset$, or*

   (b) *$n = 2$ and $\gamma \neq 1$, or*

   (c) *$n = 2$, $\gamma = 1$ and $Cycles(Y, \mathcal{G}^s\backslash\{X\}) \neq \emptyset$.*

In the remainder, we prove the above theorem through Lemmas 5.1-5.3. To do so, for the total effect $P(y_t|do(x_{t-\gamma}))$, we consider the following set:

$$\mathcal{A}_\gamma = \{(Z_{t-\gamma-\ell})_{1\leq\ell\leq\gamma_{\max}} | Z \in Desc(X; \mathcal{G}^s)\}$$
$$\cup \{(Z_{t-\gamma-\ell})_{0\leq\ell\leq\gamma_{\max}} | Z \in \mathcal{V}^s\backslash Desc(X, \mathcal{G}^s)\} \quad (2)$$

and we prove that it is a valid adjustment set when the total effect is identifiable. As one can note, it contains all possible parents of $X_{t-\gamma}$ in all candidate FTCGs of $\mathcal{G}^s$. Thus, $\mathcal{A}_\gamma$ blocks any backdoor path $\pi$ between $X_{t-\gamma}$ and $Y_t$ in any candidate FTCG through the parent of $X_{t-\gamma}$ on that path.

We first introduce the notion of ambiguous vertices, represented in gray in every figure, that will be useful for the proofs of most of the lemmas.

**Definition 6** (Ambiguous vertices). *Consider an SCG $\mathcal{G}^s$ and the total effect $P(y_t \mid do(x_{t-\gamma}))$, for $\gamma \geq 0$. A vertex $V_{t'}$ belonging to an active backdoor path for $(X_{t-\gamma}, Y_t)$ in a candidate FTCG is* ambiguous *if there exists another candidate FTCG in which $V_{t'}$ is a descendant of $X_{t-\gamma}$.*

Ambiguous vertices are crucial for identifiability. In addition to ambiguous vertices, one can also define ambiguous paths, as follows.

---

[3]In Supplementary Material, we provide an equivalent version of Theorem 2 which might be easier to read to certain readers.

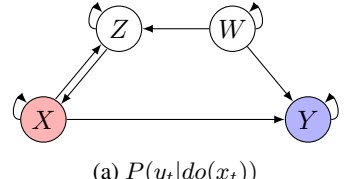
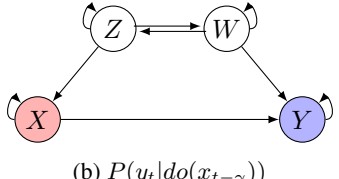
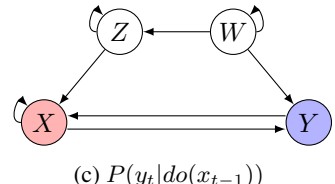

(a) $P(y_t|do(x_t))$        (b) $P(y_t|do(x_{t-\gamma}))$        (c) $P(y_t|do(x_{t-1}))$

Figure 5.1: Three SCGs and a total effect which is identifiable. Each pair of red and blue vertices in the FTCGs represents the total effect we are interested in, and we precise the total effect and the lag considered in the caption. This illustrates Lemma 5.2 (Figure a-b) and Lemma 5.3 (Figure c).

**Definition 7** (Ambiguous paths). *Consider an SCG $\mathcal{G}^s$ and a candidate FTCG $\mathcal{G}^f$. A path $\pi^f \in \mathcal{G}^f$ between $X_{t-\gamma}$ and $Y_t$, for $\gamma \geq 0$, is an* ambiguous path *if it does not contain any vertex at time $t - \gamma - \ell$ for $\ell \geq 1$. We note $\Pi_\gamma^f$ the set of all ambiguous paths in $\mathcal{G}^f$.*

When $\pi$ is not an ambiguous path ($\pi \notin \Pi_\gamma^f$), then at least one vertex on $\pi$ is in the past of $X_{t-\gamma}$ and thus cannot be ambiguous. One thus has the following property:

**Property 1.** *Consider an SCG $\mathcal{G}^s$ and the total effect $P(y_t \mid do(x_{t-\gamma}))$, for $\gamma \geq 0$. Suppose $\pi^f$ is a backdoor path between $X_{t-\gamma}$ and $Y_t$ in a candidate FTCG $\mathcal{G}^f$. If $\pi^f \notin \Pi_\gamma^f$, then $\pi^f$ is blocked by a subset of $\mathcal{A}_\gamma$ containing at least one non-ambiguous vertex.*

**Example 5.1.** *For example, in Figure 5.5c, $\pi_1^f = \langle X_{t-1}, X_{t-2}, Y_{t-1}, Y_t \rangle$ is not an ambiguous path between $X_{t-1}$ and $Y_t$ since $X_{t-2}$ precedes $X_{t-1}$ in time. On the other hand, $\pi_2^f = \langle X_{t-1}, Y_{t-1}, Y_t \rangle$ is an ambiguous path between $X_{t-1}$ and $Y_t$. The path $\pi_1^f$ is blocked by $X_{t-2}$.*

We now introduce the notion of compatible path that will allow us to relate backdoor paths in a given SCG and its candidate FTCGs.

**Definition 8** (Compatible path). *Consider an SCG $\mathcal{G}^s$, a candidate FTCG $\mathcal{G}^f$, and the total effect $P(y_t \mid do(x_{t-\gamma}))$, for $\gamma \geq 0$. We say that a path $\pi^f = \langle X_{t-\gamma}, W_{t^2}^2, \cdots, W_{t^{m-1}}^{m-1}, Y_t \rangle$ in $\mathcal{G}^f$ is* compatible *with a path $\pi^s = \langle X, V^2, \cdots, V^{n-1}, Y \rangle$ in $\mathcal{G}^s$ if for all $(W_{t^j}^j)_{2 \leq j \leq m-1}$: either $W^j \in \langle V^2, \cdots, V^{n-1} \rangle$ or $\exists V \in \langle V^2, \cdots, V^{n-1} \rangle$ such that $W^j \in Cycles(V, \mathcal{G}^s) \backslash Cycles(X, \mathcal{G}^s)$.*

The following property relates backdoor paths in a given SCG and in any of its candidate FTCG.

**Property 2.** *Consider an SCG $\mathcal{G}^s$ and the total effect $P(y_t \mid do(x_{t-\gamma}))$ for $\gamma \geq 0$. Then $(i) \Rightarrow (ii)$, where:*

*(i) $\gamma = 0$ or $Cycles^>(X, \mathcal{G}^s \backslash \{Y\}) = \emptyset$,*

*(ii) in any candidate FTCG $\mathcal{G}^f$, there exists no backdoor path $\pi^f \in \Pi_\gamma^f$ that is not compatible with any backdoor path in $\mathcal{G}^s$.*

The two above properties allow one to prove the following lemmas which prove that each condition of Theorem 2 is sufficient. The first lemma is rather straightforward and concern the case where $X \notin Anc(Y, \mathcal{G}^s)$ for a given SCG $\mathcal{G}^s$.

**Lemma 5.1.** *Consider an SCG $\mathcal{G}^s$, $\gamma \geq 0$ fixed and the total effect $P(y_t \mid do(x_{t-\gamma}))$. If $X \notin Anc(Y, \mathcal{G}^s)$ then $P(y_t \mid do(x_{t-\gamma}))$ is identifiable, and $P(y_t \mid do(x_{t-\gamma})) = P(y_t)$.*

The following lemma excludes both Conditions 1 and 2 of Theorem 2 by considering the negation of Condition 1 (in (i)) and the situation in which there is no $\sigma$-active backdoor path from $X$ to $Y$ with $\mathcal{Z} = \emptyset$.

**Lemma 5.2.** *Consider an SCG $\mathcal{G}^s$, $\gamma \geq 0$ fixed and the total effect $P(y_t \mid do(x_{t-\gamma}))$. If $X \in Anc(Y, \mathcal{G}^s)$ and*

*(i) either $\gamma = 0$ or $Cycles^>(X, \mathcal{G}^s \backslash \{Y\}) = \emptyset$ and*

*(ii) $\nexists \sigma$-active backdoor path $\pi^s = \langle V^1 = X, \cdots, V^n = Y \rangle$ from $X$ to $Y$ in $\mathcal{G}^s$ such that $\langle V^2, \cdots, V^{n-1} \rangle \subseteq Desc(X, \mathcal{G}^s)$,*

*then $P(y_t \mid do(x_{t-\gamma}))$ is identifiable by $\mathcal{A}_\gamma$.*

This lemma is illustrated in Figure 5.1a - 5.1b.

When there is a $\sigma$-active backdoor path from $X$ to $Y$ with $\mathcal{Z} = \emptyset$, the negation of Condition 2 of Theorem 2 is obtained with $n = 2$, $\gamma = 1$ and $Cycles(Y, \mathcal{G}^s \backslash \{X\}) = \emptyset$. The negation of Condition 1 of Theorem 2 is obtained in this setting with $Cycles^>(X, \mathcal{G}^s \backslash \{Y\}) = \emptyset$. Note that, as before, having a $\sigma$-active backdoor path from $X$ to $Y$ with $\mathcal{Z} = \emptyset$ and $n = 2$ is equivalent to $X \leftrightarrows Y$.

**Lemma 5.3.** *Consider an SCG $\mathcal{G}^s$ and the total effect $P(y_t \mid do(x_{t-1}))$ ($\gamma = 1$). If the only $\sigma$-active backdoor path from $X$ to $Y$ in $\mathcal{G}^s$ with $\mathcal{Z} = \emptyset$ is $X \leftrightarrows Y \in \mathcal{G}^s$ and*

*(i) $Cycles^>(X, \mathcal{G}^s \backslash \{Y\}) = \emptyset$ and*

*(ii) $Cycles(Y, \mathcal{G}^s \backslash \{X\}) = \emptyset$,*

*then $P(y_t \mid do(x_{t-1}))$ is identifiable by $\mathcal{A}_\gamma$.*

This lemma is illustrated in Figure 5.1c.

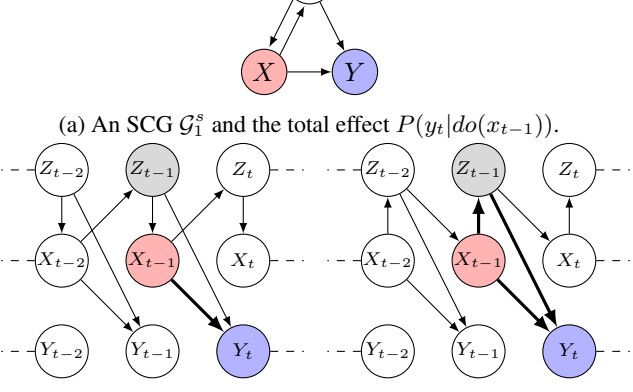

(a) An SCG $\mathcal{G}_1^s$ and the total effect $P(y_t|do(x_{t-1}))$.

(b) A first candidate FTCG.

(c) Another candidate FTCG.

Figure 5.2: An example of an SCG $\mathcal{G}_1^s$ in (a) satisfying Condition 1 in Theorem 2 and two candidate FTCGs in (b) and (c). Each pair of red and blue vertices in the FTCGs represents the total effect we are interested in. Gray vertices are ambiguous: they are on an active backdoor path in (b) and belong to a directed path in (c) (bold edges indicate direct paths from $X_{t-1}$ to $Y_t$).

## 5.2 ADJUSTMENT SET

When the total effect is identifiable and when $X \in Anc(Y, \mathcal{G}^s)$, the set $A_\gamma$ defined in Equation (2) is a valid adjustment set, but it has a large size, so we provide a smaller valid adjustment set, defined as follows:

$$\mathcal{A}'_\gamma = \{V_{t'} \in \mathcal{A}_\gamma | V \in Anc(X, \mathcal{G}^s) \cup Anc(Y, \mathcal{G}^s)\}.$$

**Proposition 2.** *Consider an SCG $\mathcal{G}^s$ and the total effect $P(y_t \mid do(x_{t-\gamma}))$, with $\gamma \geq 0$. Under conditions of identifiability provided by Theorem 2, the set $\mathcal{A}'_\gamma$ is a valid adjustment set for the total effect.*

## 5.3 NON IDENTIFIABLE EXAMPLES

In this section, we provide several examples of SCGs where the total effect cannot be identified by finding a valid adjustment set.

**Example 5.2.** *Consider the SCG in Figure 5.2a and the two candidate FCTGs given in Figure 5.2b and 5.2c. Suppose we are interested in the total effect $P(y_t \mid do(x_{t-1}))$. In the first FCTG depicted in Figure 5.2b, the path $\langle X_{t-1}, Z_{t-1}, Y_t \rangle$ is an active back-door path. Since $Z_{t-1}$ is the only vertex on this path that is not an endpoint, we need to adjust for it to eliminate the confounding bias induced by this path. However, in the second FTCG depicted in 5.2c, $\langle X_{t-1}, Z_{t-1}, Y_t \rangle$ forms a directed path. This implies that we should not adjust for $Z_{t-1}$ to preserve the*

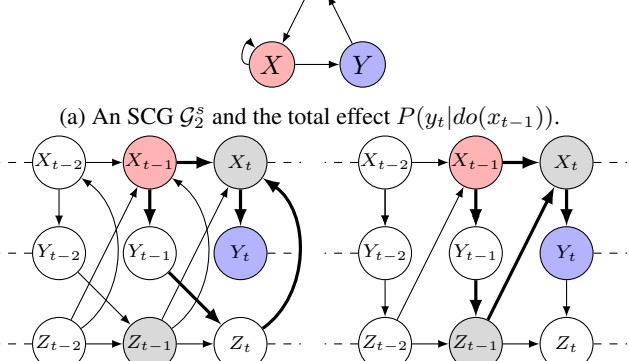

(a) An SCG $\mathcal{G}_2^s$ and the total effect $P(y_t|do(x_{t-1}))$.

(b) A first candidate FTCG.

(c) Another candidate FTCG.

Figure 5.3: An example of an SCG $\mathcal{G}_2^s$ in (a) satisfying Condition 2a in Theorem 2 and two candidate FTCGs in (b) and (c). Each pair of red and blue vertices in the FTCGs represents the total effect we are interested in. Gray vertices are ambiguous: they are on an active backdoor path in (b) and belong to a directed path in (c) (bold edges indicate direct paths from $X_{t-1}$ to $Y_t$).

*influence of $X_{t-1}$ on $Y_t$ through the path passing by $Z_{t-1}$. Since we do not know which FTCG is the true one, then we cannot determine whether we should adjust for $Z_{t-1}$ or not. Consequently, there is no valid adjustment set to identify the total effect $P(y_t \mid do(x_{t-1}))$.*

**Example 5.3.** *Consider the SCG in Figure 5.3a and the two candidate FTCGs in Figures 5.3b and 5.3c. Suppose we are interested in the total effect $P(y_t \mid do(x_{t-1}))$. The path $\langle X_{t-1}, Z_{t-1}, X_t, Y_t \rangle$ is an active back-door path in the first FTCG depicted in (b). Since $Z_{t-1}$ is the only vertex on this path that is not an endpoint and that does not belong to a directed path in the same graph, we need to adjust for it to eliminate the confounding bias induced by this path. However, in the second FTCG depicted in 5.3c, $\langle X_{t-1}, \langle Y_{t-1}, Z_{t-1}, X_t, Y_t \rangle$ forms a directed path. This implies that we should not adjust for $Z_{t-1}$ to preserve the influence of $X_{t-1}$ on $Y_t$ through the path passing by $Z_{t-1}$. Since we do not know which FTCG is the true one, then we cannot determine whether we should adjust for $Z_{t-1}$ or not. Consequently, there is no valid adjustment set to identify the total effect $P(y_t \mid do(x_{t-1}))$.*

**Example 5.4.** *Consider the SCG in Figure 5.4a and the two candidate FTCGs in Figures 5.4b and 5.4c. Suppose we are interested in the the total effect $P(y_t \mid do(x_{t-2}))$. The path $\langle X_{t-2}, Y_{t-2}, X_{t-1}, Y_t \rangle$ is an active back-door path in the first FTCG depicted in 5.4b. Since $Y_{t-2}$ is the only vertex on this path that is not an endpoint and that does not belong to a directed path in the same graph, we need to adjust for it to eliminate the confounding bias induced*

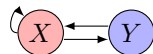

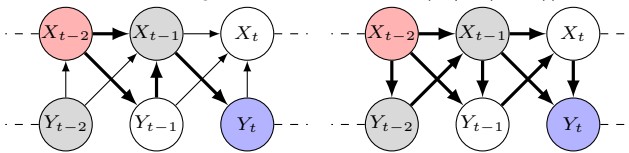

(a) An SCG $\mathcal{G}_3^s$ and the total effect $P(y_t|do(x_{t-2}))$.

(b) A first candidate FTCG.  (c) Another candidate FTCG.

Figure 5.4: An example of an SCG $\mathcal{G}_3^s$ in (a) satisfying Condition 2b in Theorem 2 with respect to $\Pr(y_t \mid do(x_{t-2}))$ and two candidate FTCGs in (b) and (c). Each pair of red and blue vertices in the FTCGs represents the total effect we are interested in. Gray vertices are ambiguous: they constitute a backdoor path in (b) and belong to a directed path in (c) (bold edges indicate direct paths from $X_{t-2}$ to $Y_t$).

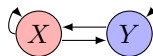

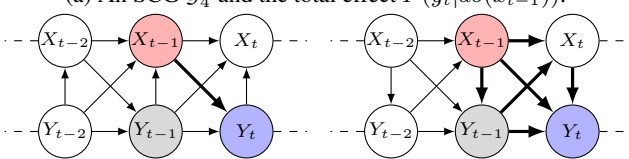

(a) An SCG $\mathcal{G}_4^s$ and the total effect $P(y_t|do(x_{t-1}))$.

(b) A first candidate FTCG.  (c) Another candidate FTCG.

Figure 5.5: An example of an SCG $\mathcal{G}_4^s$ in (a) satisfying Condition 2c in Theorem 2 with respect to $\Pr(y_t \mid do(x_{t-1}))$ and two candidate FTCGs in (b) and (c). Each pair of red and blue vertices in the FTCGs represents the total effect we are interested in. Gray vertices are ambiguous: they constitute a backdoor path in (b) and belong to a directed path in (c) (bold edges indicate direct paths from $X_{t-1}$ to $Y_t$).

*by this path. However, in the second FTCG depicted in 5.4c, $\langle X_{t-2}, Y_{t-2}, X_{t-1}, Y_t \rangle$ forms a directed path. This implies that we should not adjust for $Y_{t-2}$ to preserve the influence of $X_{t-2}$ on $Y_t$ through the path passing by $Y_{t-2}$. Since we do not know which FTCG is the true one, then we cannot determine whether we should adjust for $Y_{t-2}$ or not. Consequently, there is no valid adjustment set to identify the total effect $P(y_t \mid do(x_{t-2}))$.*

**Example 5.5.** *Consider the SCG in Figure 5.5a and the two candidate FTCGs in Figures 5.5b and 5.5c. Suppose we are interested in the the total effect $P(y_t \mid do(x_{t-1}))$. The path $\langle X_{t-1}, Y_{t-1}, Y_t \rangle$ is an active back-door path in the first FTCG depicted in 5.5b. Since $Y_{t-1}$ is the only vertex on this path that is not an endpoint, we need to adjust for it to eliminate the confounding bias induced by this path. However, in the second FTCG depicted in 5.5c, $\langle X_{t-1}, Y_{t-1}, Y_t \rangle$ forms a directed path. This implies that we should not adjust for $Y_{t-1}$ to preserve the influence of $X_{t-1}$ on $Y_t$ through the path passing by $Y_{t-1}$. Since we do not know which FTCG is the true one, then we cannot determine whether we should adjust for $Y_{t-1}$ or not. Consequently, there is no valid adjustment set to identify the total effect $P(y_t \mid do(x_{t-1}))$.*

Notice that in Figure 5.5, removing the self-loop on $Y$ makes the total effect identifiable. This is because the active backdoor path and the directed path discussed in Example 5.5 would no longer exist, leaving only directed paths or blocked (due to a collider) backdoor paths between $X_{t-1}$ and $Y_t$.

# 6 DISCUSSION ON REAL-WORLD APPLICATIONS

**Nephrology.** Hypertension has long been considered as a risk factor for kidney function decline. At the same time, the kidney is known to have a major role in affecting blood pressure through sodium extraction and regulating electrolyte balance [Yu et al., 2020]. This can be represented with the SCG in Figure 6.1a where the kidney function is represented by the creatinine level. Epidemiologists are interested to know if preventing kidney function decline can reduce the public health burden of hypertension and at the same time nephrologists are interested in knowing how much a treatment related to hypertension can improve the state of the kidney. Using Theorem 2 and assuming no hidden confounding, we can identify the total effect in each direction with a lag equal to 1 (if there are confounders that do not form additional cycles, the total effect remains identifiable if we measure them and take them into account in the SCG). We can collect data for estimation by conducting weekly blood tests on patients with kidney insufficiency, especially those whose hypertension and creatinine levels fluctuate.

**Finance.** It has been suggested that there exists a bidirectional causal relationship between the number of unique active wallets associated with bridge protocols and the mean transaction fees within the Ethereum network [Ante and Saggu, 2024]. Additionally, we consider that transaction fees causes itself over time, as depicted in the SCG shown in Figure 6.1b. In this scenario, the total effect of mean transaction fees on the number of unique active wallets is identifiable using Theorem 2 with a lag of 1. However, the same does not hold true for the opposite direction: the total effect of the number of unique active wallets on the mean transaction fees is not identifiable using Theorem 2.

**System monitoring.** Consider a subgraph of the SCG described in [Bystrova et al., 2024], representing the web activity in an IT system. Suppose that system experts observed a high number of queries at midnight for several weeks, likely due to a Distributed Denial of Service attack. Simultaneously, they noticed that CPU usage at midnight was very high, preventing the system from running some processes. Therefore, the system experts would like to determine (before intervening in the system) how much a reduction in

bandwidth in the network would reduce the global CPU usage. Theorem 2 shows that the total effect of Network input on CPU Global is identifiable for any lag. In addition, Theorem 2 implies that the total effect between all pairs of variables is identifiable since in the SCG there exists no cycles of size greater than 2. We can estimate those total effect using the data introduced in Bystrova et al. [2024].

**Thermoregulation.** Inspired by the experiment conducted in Peters et al. [2013], we consider maintaining a steady temperature in an apartment composed of four rooms: a living room, a kitchen, a bathroom, and an office. The living room is the only room containing a radiator, and all rooms are connected to each other through the living room. Additionally, all rooms contain a window except for the office. Temperature sensors were placed in the four rooms, plus one outside the apartment, and temperatures were recorded on an hourly basis. We consider the SCG presented in Figure 6.1d as the true one. Clearly, the outside temperature directly influences all rooms containing a window and the temperature in each room cannot cause the outside temperature. Since the living room contains a radiator, it can affect the temperatures in all other rooms. Additionally, since we may use fire in the kitchen for cooking, which can increase the temperature, we consider that the temperature in the kitchen can affect the temperature in the living room. Similarly, since we may use hot water in the bathroom, which can increase the temperature, we consider that the temperature in the bathroom can influence the temperature in the living room. All other vertices representing rooms in the graph are not connected to each other because they are not physically directly connected; they are all connected through the living room. Suppose we are specifically interested in estimating the total effect of the temperature in living room on the temperature in the office. Theorem 2 states that this total effect is identifiable for any lag since $Cycles(\text{Living Room}, \mathcal{G}\backslash\{\text{Office}\}) = \emptyset$ and there exists no $\sigma$-active backdoor path between Living Room and Office.

# 7 CONCLUSION

We studied in this paper the identification of total effects between singleton variables, under causal sufficiency, for both extended summary causal graphs and summary causal graphs. We showed that the total effect is always identifiable for extended summary causal graphs. The same does not hold for summary causal graphs for which we established graphical conditions which are sufficient, in any underlying probability distribution, for the identifiability of the total effect. In addition, in case of identifiability, we provided several valid adjustment sets for estimating the total effect in extended summary causal graphs, and two adjustment sets when considering summary causal graphs.

These results have significant implications, such as impact analysis in dynamic systems, particularly in scenarios where

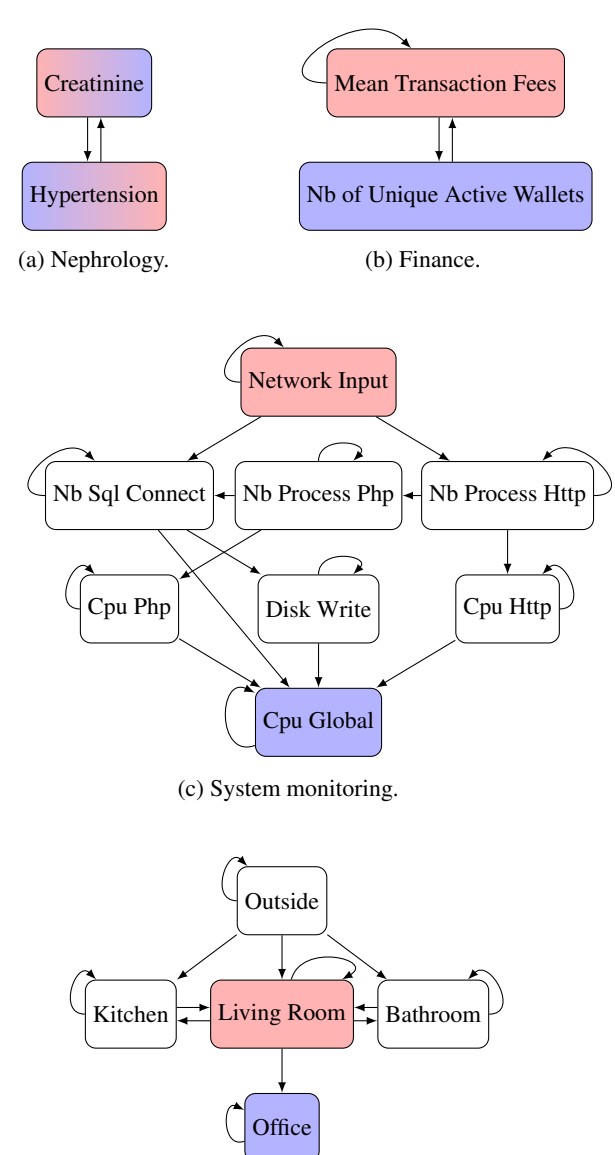

(a) Nephrology.

(b) Finance.

(c) System monitoring.

(d) Thermoregulation.

Figure 6.1: Real summary causal graphs from (a) Nephrology, (b) Finance, (c) System Monitoring, and (d) Thermoregulation. Each pair of red and blue vertices represents the total effect(s) of interest where the red vertex represents the intervention and the blue vertex represents the response. According to Theorem 2, each of these total effects is either identifiable in general or identifiable under certain conditions on $\gamma$.

experts are unable to provide either a full temporal causal graph or an extended summary causal graph. They are also valuable in cases where the assumptions underlying causal discovery methods for inferring causal graphs with time lags are deemed overly restrictive. Furthermore, these results offer insights that can be useful in different disciplines such as Nephrology, Finance, System Monitoring, and Thermoregulation.

For future works, it would be valuable to establish necessary and sufficient conditions for the identifiability of total effects using SCGs, to extend this work to the case where the responses and interventions can be multivariate, and to the case where there are hidden confounding.

## Acknowledgements

We thank Ali Aït-Bachir from EasyVista for discussions about the application of this work in system monitoring. We thank Benjamin Glemain and Nathanael Lapidus from IPLESP and Paolo Malvezzi from CHU Grenoble for discussions about the application of this work in Nephrology/Epidemiology. Finally, we thank Clément Yvernes from LIG for several discussions and five anonymous reviewers for their many insightful comments and suggestions. This work was partially supported by MIAI@Grenoble Alpes (ANR-19-P3IA-0003), by the CIPHOD project (ANR-23-CPJ1-0212-01), and by the CSPR R&D Booster Auvergne-Rhône-Alpes project.

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

# A  SUPPLEMENTARY MATERIAL

## A.1  PROOFS OF SECTION 4

**Theorem 1.** *(Identifiability in ESCG) Consider an ESCG $\mathcal{G}^e$. Under Assumptions 1 and 2 for $\mathcal{G}^e$, the total effect $P(y_t|do(x_{t-\gamma}))$ is identifiable in $\mathcal{G}^e$ for any $\gamma \geq 0$. Furthermore, the set*

$$\mathcal{B}_\gamma = \{(Z_{t-\gamma-\ell})_{1 \leq \ell \leq \gamma_{\max}} | Z_{t^-} \in Par(X_t, \mathcal{G}^e)\}$$
$$\cup \{Z_{t-\gamma} | Z_t \in Par(X_t, \mathcal{G}^e)\},$$

*is a valid adjustment set for $P(y_t|do(x_{t-\gamma}))$ for the formulae given in Equation (1).*

*Proof.* If $X \notin Anc(Y, \mathcal{G}^e)$, then in every candidate FTCG $\mathcal{G}^f$, $X_{t-\gamma} \notin Anc(Y_t, \mathcal{G}^f)$. Thus, $P(y_t \mid do(x_{t-\gamma}))$ is always identifiable in $\mathcal{G}^e$, and $P(y_t \mid do(x_{t-\gamma})) = P(y_t)$.

Assume now that $X \in Anc(Y, \mathcal{G}^e)$. Let $\gamma_{\max}$ be the maximal lag, and $\mathcal{G}^f$ be a candidate FTCG. We prove that

$$\mathcal{B}_\gamma = \{(Z_{t-\gamma-\ell})_{1 \leq \ell \leq \gamma_{\max}} | Z_{t^-} \in Par(X_t, \mathcal{G}^e)\}$$
$$\cup \{Z_{t-\gamma} | Z_t \in Par(X_t, \mathcal{G}^e)\}$$

is an adjustment set for $P(y_t|do(x_{t-\gamma}))$ in $\mathcal{G}^f$.

First, we have to prove that $Par(X_{t-\gamma}, \mathcal{G}^f) \subseteq \mathcal{B}_\gamma$. Let $Z_{t-\gamma-\ell} \in Par(X_{t-\gamma}, \mathcal{G}^f)$. If $\ell = 0$, then $Z_t$ causes $X_t$ in $\mathcal{G}^e$ by consistency throughout time, which means that $Z_{t-\gamma} \in \mathcal{B}_\gamma$. If $\ell > 0$, then $Z_{t^-}$ causes $X_t$ in $\mathcal{G}^e$, that is $Z_{t-\ell-\gamma} \in \mathcal{B}_\gamma$. This shows that the set $\mathcal{B}_\gamma$ blocks all backdoor paths relatively to $P(y_t|do(x_{t-\gamma}))$.

Then, we have to prove $\mathcal{B}_\gamma$ does not contain any descendant of $X_{t-\gamma}$ in $\mathcal{G}^f$. If this is true, there exists $W_{t-\gamma} \in \mathcal{B}_\gamma \cap Desc(X_{t-\gamma}, \mathcal{G}^f)$, at time slice $t - \gamma$ because it is a parent and a descendant of $X_{t-\gamma}$. By consistency throughout time, $W_t \in Desc(X_t, G^f)$. However, by definition of $\mathcal{B}_\gamma$, $X_t \in Desc(W_t, G^f)$, which contradicts the acyclicity assumption of $\mathcal{G}^e$. It means that neither it blocks any directed path between $X_{t-\gamma}$ and $Y_t$, nor it contains any descendant of $Y_t$. □

**Proposition 1.** *Consider an ESCG $\mathcal{G}^e$ and a maximal lag $\gamma_{\max}$ and let $\gamma \geq 0$. Any adjustment set $\mathcal{B}'_\gamma$ for the total effect $P(y_t|do(x_{t-\gamma}))$ that satisfies the standard backdoor criterion on the densest candidate FTCG in $\mathcal{C}(\mathcal{G}^e)$ is a valid adjustment set for the total effect. In addition, $\mathcal{B}_\gamma$ is a valid adjustment set with respect to the standard backdoor criterion on the densest candidate FTCG.*

*Proof.* Let $\mathcal{G}^f_d$ be the densest candidate FTCG, and $\mathcal{B}'_\gamma$ be an adjustment set in $\mathcal{G}^f_d$. Let $\mathcal{G}^f$ be another candidate FTCG. By definition of $\mathcal{G}^f_d$, any back-door path in $\mathcal{G}^f$ is also in $\mathcal{G}^f_d$ (the last graph contains all possible edges). Then, $\mathcal{B}'_\gamma$ blocks all back-door paths in $\mathcal{G}^f$. Moreover, since no vertex in $\mathcal{B}'_\gamma$ is a descendant of $X_{t-\gamma}$ in $\mathcal{G}^f_d$, the same holds for $\mathcal{G}^f$. Thus, $\mathcal{B}'_\gamma$ is also an adjustment set in $\mathcal{G}^f$. $\qquad\square$

## A.2 PROOFS OF SECTION 5

**Property 1.** *Consider an SCG $\mathcal{G}^s$ and the total effect $P(y_t \mid do(x_{t-\gamma}))$, for $\gamma \geq 0$. Suppose $\pi^f$ is a backdoor path between $X_{t-\gamma}$ and $Y_t$ in a candidate FTCG $\mathcal{G}^f$. If $\pi^f \notin \Pi^f_\gamma$, then $\pi^f$ is blocked by a subset of $\mathcal{A}_\gamma$ containing at least one non-ambiguous vertex.*

*Proof.* Suppose $\pi^f$ is path between $X_{t-\gamma}$ and $Y_t$ for $\gamma \geq 0$. If $\pi^f \notin \Pi^f_\gamma$ and then $\pi^f$ contains at least one vertex $Z_{t-\gamma-\ell}$ for $\ell \geq 1$. $Z_{t-\gamma-\ell}$ is temporally prior to $X_{t-\gamma}$ which means $\pi^f$ that if $\pi^f$ is a backdoor path then adjusting on $Z_{t-\gamma-\ell}$ and the parents of $Z_{t-\gamma-\ell}$ on the path will block the path. Furthermore, for the same reason, there cannot be a directed path from $X_{t-\gamma}$ to $Z_{t-\gamma-\ell}$ in any FTCG. Finally, again for the same reason, $Z_{t-\gamma-\ell}$ and the parents of $Z_{t-\gamma-\ell}$ are in $\mathcal{A}_\gamma$. $\qquad\square$

**Property 2.** *Consider an SCG $\mathcal{G}^s$ and the total effect $P(y_t \mid do(x_{t-\gamma}))$ for $\gamma \geq 0$. Then $(i) \Rightarrow (ii)$, where:*

(i) *$\gamma = 0$ or $Cycles^>(X, \mathcal{G}^s\backslash\{Y\}) = \emptyset$,*

(ii) *in any candidate FTCG $\mathcal{G}^f$, there exists no backdoor path $\pi^f \in \Pi^f_\gamma$ that is not compatible with any backdoor path in $\mathcal{G}^s$.*

*Proof.* Assume first $Cycles^>(X, \mathcal{G}^s\backslash\{Y\}) = \emptyset$. Suppose $\exists \pi^f = X_{t-\gamma} \leftarrow W_{t-\gamma} \cdots \rightarrow Y_t \in \Pi^f_\gamma$ which is a backdoor path between $X_{t-\gamma}$ and $Y_t$ that is not compatible with any back-door path $\pi^s = \langle V^1 = X, V^2, \cdots, V^{n-1}, V^n = Y\rangle$ in $\mathcal{G}^s$.

If $n = 2$, then the path compatible with the cycle $\langle X, X\rangle$ is of the form $X_{t-\gamma} \rightarrow X_{t-\gamma+i} \rightarrow \cdots \rightarrow X_{t-\gamma+j} \rightarrow Y_t$: it means that $\pi^f$ cannot be a back-door path.

If $n > 2$, $W_{t-\gamma}$ is such that $W \notin \{V^2, \cdots, V^{n-1}\}$ and $\nexists V \in \{V^2, \cdots, V^{n-1}\}$ such that $W \in Cycles(V, \mathcal{G}^s)$. If the path between $W_{t-\gamma}$ and $Y_t$ does not pass by $X_{t-\gamma+\ell}$

with $\ell > 0$, then there exists a back-door path between $X$ and $Y$ passing by $W$ in $\mathcal{G}^s$ as $\pi^f$ lies in a candidate FTCG, which contradicts our assumption. So the path necessarily passes by $X_{t-\gamma+\ell}$. Thus there is a cycle $C_x$ on $X$ such that $size(C_x) > 2$, which leads again to a contradiction. Thus, there does not exist a back-door path $\pi^f \in \Pi^f_\gamma$ between $X_{t-\gamma}$ and $Y_t$ that is not compatible with any back-door path in $\mathcal{G}^s$.

The case $\gamma = 0$ is treated in the same way, with the fact that the path considered cannot go back to $X_t$ as this would create a cycle in the FTCG. $\qquad\square$

**Lemma 5.1.** *Consider an SCG $\mathcal{G}^s$, $\gamma \geq 0$ fixed and the total effect $P(y_t \mid do(x_{t-\gamma}))$. If $X \notin Anc(Y, \mathcal{G}^s)$ then $P(y_t \mid do(x_{t-\gamma}))$ is identifiable, and $P(y_t \mid do(x_{t-\gamma})) = P(y_t)$.*

*Proof.* If $X \notin Anc(Y, \mathcal{G}^s)$, then in every candidate FTCG $\mathcal{G}^f$, $X_{t-\gamma} \notin Anc(Y_t, \mathcal{G}^f)$. Thus, $P(y_t \mid do(x_{t-\gamma}))$ is always identifiable in $\mathcal{G}^s$, and $P(y_t \mid do(x_{t-\gamma})) = P(y_t)$. $\qquad\square$

**Lemma 5.2.** *Consider an SCG $\mathcal{G}^s$, $\gamma \geq 0$ fixed and the total effect $P(y_t \mid do(x_{t-\gamma}))$. If $X \in Anc(Y, \mathcal{G}^s)$ and*

(i) *either $\gamma = 0$ or $Cycles^>(X, \mathcal{G}^s\backslash\{Y\}) = \emptyset$ and*

(ii) *$\nexists \sigma$-active backdoor path $\pi^s = \langle V^1 = X, \cdots, V^n = Y\rangle$ from $X$ to $Y$ in $\mathcal{G}^s$ such that $\langle V^2, \cdots, V^{n-1}\rangle \subseteq Desc(X, \mathcal{G}^s)$,*

*then $P(y_t \mid do(x_{t-\gamma}))$ is identifiable by $\mathcal{A}_\gamma$.*

*Proof.* We will prove that $A_\gamma$ is an adjustment set for $P(y_t \mid do(x_{t-\gamma}))$ in any candidate FTCG under conditions (i) and (ii). Let $\mathcal{G}^f$ be a candidate FTCG, and $\Pi^f_\gamma$ the set of ambiguous paths. By Property 1, any back-door path $\pi^f \notin \Pi^f_\gamma$ can be blocked by $\mathcal{A}_\gamma$. Furthermore, by definition, elements of $\mathcal{A}_\gamma$ cannot be descendant of $X_{t-\gamma}$.

We now turn our attention to paths in $\Pi^f_\gamma$. Let $\pi^f \in \Pi^f_\gamma$ be a back-door path between $X_{t-\gamma}$ and $Y_t$. Since $\gamma = 0$ or $Cycles^>(X, \mathcal{G}^s\backslash\{Y\}) = \emptyset$ then by Property 2, all back-door paths in $\Pi^f_\gamma$ are compatible with back-door paths in $\mathcal{G}^s$. Let $\pi^s = \langle V^1 = X, \cdots, V^n = Y\rangle$ be a $\sigma$-active back-door path in $\mathcal{G}^s$ compatible with $\pi^f$. By (ii), there exists $m \geq 1$ vertices such that $\{V^{i_1}, \cdots, V^{i_m}\} \subseteq \langle V^2, \cdots, V^{n-1}\rangle$ and $\{V^{i_1}, \cdots, V^{i_m}\} \not\subset Desc(X, \mathcal{G}^s)$. Then, $\forall V_{t-\gamma}$ such that $V \in \{V^{i_1}, \cdots, V^{i_m}\}$, $V_{t-\gamma} \notin Desc(X_{t-\gamma}, \mathcal{G}^f)$ and since $X \in Anc(Y, \mathcal{G}^s)$ then it must be the case that $V \notin Desc(Y, \mathcal{G}^s)$ and by consequence $V_{t-\gamma} \notin Desc(Y_t, \mathcal{G}^f)$. Thus, $V_{t-\gamma}$ cannot be an ambiguous vertex. Its parent in $\pi^f$ furthermore blocks $\pi^f$, is not ambiguous (as otherwise $V_{t-\gamma}$ would be ambiguous) and is a member of $\mathcal{A}_\gamma$ by definition of $\mathcal{A}_\gamma$. Thus $\mathcal{A}_\gamma$ blocks all back-door paths between $X_{t-\gamma}$ and $Y_t$ in any candidate FTCG $\mathcal{G}^f$. Furthermore, no node in $\mathcal{A}_\gamma$ can block a directed path between $X_{t-\gamma}$ and $Y_t$ or is a descendant of $Y_t$ as nodes in $\mathcal{A}_\gamma$ are either defined before

$t - \gamma$ or are not descendant of $X_{t-\gamma}$, and thus of $Y_t$. This concludes the proof. $\qquad\square$

**Lemma 5.3.** *Consider an SCG $\mathcal{G}^s$ and the total effect $P(y_t \mid do(x_{t-1}))$ ($\gamma = 1$). If the only $\sigma$-active backdoor path from $X$ to $Y$ in $\mathcal{G}^s$ with $\mathcal{Z} = \emptyset$ is $X \leftrightarrows Y \in \mathcal{G}^s$ and*

(i) $Cycles^{>}(X, \mathcal{G}^s \backslash \{Y\}) = \emptyset$ *and*

(ii) $Cycles(Y, \mathcal{G}^s \backslash \{X\}) = \emptyset$,

*then $P(y_t \mid do(x_{t-1}))$ is identifiable by $\mathcal{A}_\gamma$.*

*Proof.* We will prove that $\mathcal{A}_1$ is an adjustment set for $P(y_t \mid do(x_{t-1}))$ in any candidate FTCG under conditions (i) and (ii). Let $\mathcal{G}^f$ be a candidate FTCG, and $\Pi_1^f$ the set of ambiguous paths.

Since (ii) then by Property 2, all back-door paths in $\Pi_1^f$ are compatible with back-door paths in $\mathcal{G}^s$. In addition, by Property 1, any path $\pi^f \notin \Pi_1^f$ can be blocked by $\mathcal{A}_1$. Therefore, in the following, we focus on paths in $\Pi_1^f$ compatible with back-door paths in $\mathcal{G}^s$.

Consider the $\sigma$-active back-door path $\pi^s = \langle X, Y \rangle$. As there cannot be a loop on $Y$ by (i), the only path $\pi^f \in \Pi_1^f$ from $X_{t-1}$ to $Y_t$ compatible with $\pi^s$ that pass by $Y_{t-1}$ is $\pi_f = \langle X_{t-1}, Y_{t-1}, X_t, Y_t \rangle$. Then, under consistency throughout time, acyclicity and temporal priority, the only choices are $X_{t-1} \to Y_{t-1} \to X_t \to Y_t$ and $X_{t-1} \leftarrow Y_{t-1} \to X_t \leftarrow Y_t$. The first is a directed path, the second a back-door path already blocked due to the collider $Y_{t-1} \to X_t \leftarrow Y_t$. Thus, all potential back-door paths between $X_{t-1}$ and $Y_t$ in any candidate $\mathcal{G}^f$ are blocked, and $\mathcal{A}_1$ does not activate them. $\qquad\square$

**Theorem 2.** *(Identifiability in SCG) Consider an SCG $\mathcal{G}^s = (\mathcal{V}^s, \mathcal{E}^s)$ associated with a DSCM with true (unknown) probability distribution $P$. Under Assumptions 1 and 2, the total effect $P(y_t | do(x_{t-\gamma}))$, with $\gamma \geq 0$, is identifiable if $X \notin Anc(Y, \mathcal{G}^s)$ or $X \in Anc(Y, \mathcal{G}^s)$ and none of the following holds:*

1. $\gamma \neq 0$ *and* $Cycles^{>}(X, \mathcal{G}^s \backslash \{Y\}) \neq \emptyset$, *or*

2. *there exists a $\sigma$-active backdoor path*

$$\pi^s = \langle V^1 = X, \cdots, V^n = Y \rangle$$

*from $X$ to $Y$ in $\mathcal{G}^s$ such that $\langle V^2, \cdots, V^{n-1} \rangle \subseteq Desc(X, \mathcal{G}^s)$ and one of the following holds:*

    (a) $n > 2$, *i.e.* $\langle V^2, \cdots, V^{n-1} \rangle \neq \emptyset$, *or*

    (b) $n = 2$ *and* $\gamma \neq 1$, *or*

    (c) $n = 2$, $\gamma = 1$ *and* $Cycles(Y, \mathcal{G}^s \backslash \{X\}) \neq \emptyset$.

*Proof.* The proof of this theorem is given by Lemmas 5.1-5.3. $\qquad\square$

**Proposition 2.** *Consider an SCG $\mathcal{G}^s$ and the total effect $P(y_t \mid do(x_{t-\gamma}))$, with $\gamma \geq 0$. Under conditions of identifiability provided by Theorem 2, the set $\mathcal{A}'_\gamma$ is a valid adjustment set for the total effect.*

*Proof.* Let $\mathcal{G}^f$ be a candidate FTCG. Consider $V_{t'} \in \mathcal{A}_\gamma \backslash \mathcal{A}'_\gamma$: by definition of $\mathcal{A}'_\gamma$, it follows that $V_{t'} \notin Anc(X_{t-\gamma}, \mathcal{G}^f) \cup Anc(Y_t, \mathcal{G}^f)$. Therefore $V_{t'}$ does not lie on any back-door path between $X_{t-\gamma}$ and $Y_t$: $V_{t'}$ is not necessary in the adjustment set, confirming that $\mathcal{A}'_\gamma$ is also an adjustment set. $\qquad\square$

## A.3 ANOTHER VERSION OF THEOREM 2

**Theorem 2 - Version 2.** *Consider an SCG $\mathcal{G}^s = (\mathcal{V}^s, \mathcal{E}^s)$. The total effect $P(y_t \mid x_{t-\gamma})$ with $\gamma$ is identifiable from $\mathcal{G}^s$ if $X \notin Anc(Y, \mathcal{G}^s)$ or $X \in Anc(Y, \mathcal{G}^s)$ and one of the following conditions holds:*

1. $Cycles^{>}(X, \mathcal{G}^s \backslash \{Y\}) = \emptyset$ *and there exists no $\sigma$-active backdoor path $\pi^s = \langle V^1 = X, \ldots, V^n = Y \rangle$ from $X$ to $Y$ in $\mathcal{G}^s$ such that $\langle V^2, \ldots, V^{n-1} \rangle \subseteq Desc(X, \mathcal{G}^s)$ or*

2. $\gamma = 0$ *and there exists no $\sigma$-active backdoor path $\pi^s = \langle V^1 = X, \ldots, V^n = Y \rangle$ from $X$ to $Y$ in $\mathcal{G}^s$ such that $\langle V^2, \ldots, V^{n-1} \rangle \subseteq Desc(X, \mathcal{G}^s)$ or*

3. $Cycles^{>}(X, \mathcal{G}^s \backslash \{Y\}) = \emptyset$ *and there exists a $\sigma$-active backdoor path $\pi^s = \langle V^1 = X, \ldots, V^n = Y \rangle$ from $X$ to $Y$ in $\mathcal{G}^s$ such that $\langle V^2, \ldots, V^{n-1} \rangle \subseteq Desc(X, \mathcal{G}^s)$, and $n = 2$, and $\gamma = 1$, and $Cycles(Y, \mathcal{G}^s \backslash \{X\}) = \emptyset$.*