# OpenReview forum: "Identifiability of total effects from abstractions of time series causal graphs"
_auai.org/UAI/2024/Conference — UAI 2024 poster_

### Official Review · Reviewer_fkvr · 2024-03-14

**Q2-1 Originality-Novelty:** 3
**Q2-2 Correctness-Technical Quality:** 2
**Q2-5 Clarity Of Writing:** 3

**Q10 Ethical Concerns:**

No.

**Q1 Summary And Contributions:**

The paper discusses the identifiability of causal effects in a dynamic structural causal model given one of two types of abstracted information: either a summary causal graph or an extended summary causal graph. With the more informative extended summary causal graph, the effect is always identifiable, i.e., one can read off valid adjustment sets. For the summary causal graph, they provide sufficient graphical criteria to determine identifiability. Under an additional assumption on the process, these conditions become necessary.

**Q2-3 Extent To Which Claims Are Supported By Evidence:**

3: Good: the main claims are supported by convincing evidence (in the form of adequate experimental evaluation, proofs, (pseudo-)code, references, assumptions).

**Q2-4 Reproducibility:**

2: Fair: key resources (e.g. proofs, code, data) are unavailable but key details (e.g. proof sketches, experimental setup) are sufficiently well-described for an expert to confidently reproduce the main results.

**Q3 Main Strengths:**

The paper provides novel identifiability results which were rather surprising to me.
The main theorem is broken down into several subcases which are mostly supported by illustrative examples. Therefore, it is possible to understand the main theorem to a large extent without going to the detailed proofs.
The structure of the paper appears to be reasonable.
The technical results appear to be correct. (I did not read the proofs in the appendix).

**Q4 Main Weakness:**

1. The illustrative examples do not directly give intuition as to why this positive/ negative result expands to a whole class of cases. Here, some guidelines on how to construct examples of the individual Lemmata could be helpful.

2. Several times concepts that are not necessary to grasp the main result are introduced, for example, Compatibility (Definition 4) or the full definition of $\sigma$-blocked paths (Definition 7).

3. I do not understand why Assumption 3, i.e., effective intervention, should be true. Consider the simple case of $\gamma_{max}=0$, i.e., i.i.d. data. In a structure $Z \leftarrow X \rightarrow Y$, it is inefficient but not inconsistent to include $Z$ in the adjustment set. Hence, it might be better to have a separate result with the sufficient conditions for identifiability that holds regardless of this assumption.

**Q5 Detailed Comments To The Authors:**

- p2: In the section on graph notation it is unclear to me whether one implicitly restricts to directed graphs or not. All the examples use directed edges only. What defines a backdoor path if we allow for bidirected edges as well? Also, I do not understand what counterexamples to the definite status could exist, are these just bidirected edges?

- p2: By the d-separation criteria, I assume that a variable counts as its own descendant in your notation. As there is not a consensus on that, this could be mentioned.

- p2: "if and only if X causes Y at time t with a time
lag of $\gamma$''. I would argue that in $X \rightarrow Z \rightarrow Y$, $X$ causes $Y$ as well. Hence, there is not a 1 to 1 correspondence between ``causing'' and edges in the FTCG graph. I suggest framing this differently.

- p2: "if all the causal
relationships remain constant in direction throughout time.'' Can they change in magnitude? What does even define the causal effect if it is temporarily varying?

- p3: Compatibility is defined. Is this used for any of the forthcoming results? If not, maybe it could be omitted in this context.

- p4: In Definition 6, $\mathcal{Z}$ is defined as a subset of the vertices in the (E)SCG. How can it block paths in the FTCG? The vertices are not the same.

- p4: "which contains descendants of $X_{t-\gamma}$ and/or does not block all backdoor paths''. The and/or is a bit imprecise. Somewhere in the Appendix you use ``Option A or Option B or both'' in a similar case. This is less ambiguous.

- p4: The expectations used in Assumption 3 should be defined precisely. I don't understand why having descendants of $X_{t-\gamma}$ in $\mathcal{Z}$ is an issue as long as they are not mediating a directed path from $X_{t-\gamma}$ to $Y_t$.

- p5: ``If $B_\gamma$ is a valid adjustment set,'' Why 'if'? You state above that it is.

- p5: ``that form a collider at Z'' Should it be at $W$?

- p5: Colliders are defined as $\rightarrow W \leftarrow$. Aren't there more options with bidirected edges?

- p5: In Theorem 3, only $\sigma$-active backdoor paths with empty $\mathcal{Z}$ are relevant, i.e., only one bullet point could apply. If I do not miss something here, I suggest avoiding the burden of going through the complete definition of $\sigma$-blocked and restricting it to the relevant part.

- p5: There is a reminder on the definition of $Cycles()$. As this was not used before, you could just introduce it here.

- p5: You define sufficient conditions for identifiability which are necessary under Assumption 3, right? As Assumption 3 is (also) untestable, it might be worthwhile mentioning explicitly the result just with the sufficient conditions as well. In the end, this is what one is practically interested in.

- p6: Maybe some intuitive explanation of Property 1 could be of use. If there is a non-ambiguous vertex, we can condition on that to block the path. But with an ambiguous vertex, this might go wrong as we could block the wrong effects.

- p8: The caption of Figure 5.5 is contradictory. "in the FTCGs'', this should be 'SCGs' I think. ``This illustrates
Lemma 5.2 (Figure a-b) and Lemma 5.3 (Figure c).'' This should be 5.6 and 5.7.

- p8: Lemma 5.5. is almost a tautology.

- p8: Alongside these illustrative examples, providing $A_\gamma$ and $A'_\gamma$ for the identifiable cases would be helpful.

**Q9 Complying With Reviewing Instructions:**

Yes

---

> ### Author Rebuttal · Authors · 2024-04-05
>
> The reviewer's feedback and comments are deeply valued, and we have addressed all concerns raised, respecting their order of appearance.
>
> Q4 We will focus on the main contribution of our work: providing sufficient conditions for the total effect to be identifiable in SCGs. In consequence, we will streamline Section 3 by removing Definition 4, Assumption 3 and Theorem 1. Additionally, we will simplify Section 5.1. Moreover, we plan to interchange Sections 5.1 and 5.2. This involves presenting the sufficient conditions for identifiability in the revised Section 5.1, followed by a discussion of some non-identifiable cases in Section 5.2.
>
> Q5.1 We wanted to provide these definitions for both DAGs and SCGs, but we acknowledge that it wasn't clear. We will make it explicit that in an SCG, a backdoor path is defined as a path that starts with <- or <-> (we initially assumed in the submitted version that it would be clear that <- is included in <->).
>
> Q5.2 That is correct. We will explicitly state it in the text.
>
> Q5.3 We agree. We will clarify here that by a cause we mean a direct cause with respect to the graph.
>
> Q5.4 There is a typo/error in this assumption, and thank you for bringing it to our attention. It should specify that the causal relation should be constant in both direction and magnitude. Otherwise, the task of finding a unique total effect becomes ill-defined.
>
> Q5.5 We agree. We will omit it as stated in our answer to Q4.
>
> Q5.6 We acknowledge that this definition is unclear and contains typos. The set Z (we will call it for now on Zf) that is in the items is intended to be a subset of nodes in the FTCG (not in the SCG or the ESCG) that is compatible (for example, Zf=(Zt-2, Zt-1, Zt)) with the subset Z introduced in the beginning of the definition. Perhaps it would be clearer to refer to this definition as the common backdoor criterion, as it primarily focuses on FTCGs. However, it also has implications for SCGs and ESCGs: if we can identify a common set that satisfies the backdoor criterion for any possible FTCG, then it might be possible to identify this set without having any FTCG and solely by examining the SCG or the ESCG. We will clarify this point, and to simplify matters, we will concentrate on its implications for SCGs.
>
> Q5.7 It should be an « or ». But in any case we will remove Assumption 3 as stated before.
>
> Q5.8 We will remove Assumption 3 as stated before.
>
> Q5.9 Thank you for this remark. Our sentence is not clear and we will replace it by « B_γ is indeed a valid adjustment set, but at times, it may be excessively large, rendering it unsuitable for practical use. »
>
> Q5.10 Yes, thank you for pointing it out.
>
> Q5.11 We totally agree, we will redefine sigma blocked path only in the case when Z is empty.
>
> Q5.12 We agree. We will take this into account.
>
> Q5.13 Same answer as in Q4.
>
> Q5.14 Exactly! We will add your sentence to the text. Thank you!
>
> Q5.15 Thank you for pointing out the typos in Figure 5.5, we will correct them.
>
> Q5.17 True but we think it can be helpful for some readers to clearly state it in a lemma.
>
> Q5.16 We will provide Agamma and Agamma’ in each example.

---

### Official Review · Reviewer_DjMt · 2024-03-17

**Q2-1 Originality-Novelty:** 3
**Q2-2 Correctness-Technical Quality:** 2
**Q2-5 Clarity Of Writing:** 2

**Q1 Summary And Contributions:**

The paper studies the identifiability of total effects on two abstractions of the causal graphs for time series. In particular, it considers the Extended Summary Causal Graph (ESCG) and the Summary Causal Graph (SCG), which are abstractions for the Full-time Causal Graph (FTCG). Since different FTCGs (with different total effect values) may be converted into a same ESCG/SCG, the total effect may not be identifiable by just looking at the abstractions. The goal of the paper is to determine the criteria (on ESCG/SCG) under which a total effect is identifiable. With some assumptions (causal sufficiency, consistency throughout time, and effective intervention), the paper proves that the total effects are always identifiable on ESCGs and shows both necessary and sufficient conditions for testing the identifiability on SCGs.

**Q2-3 Extent To Which Claims Are Supported By Evidence:**

2: Fair: the main claims are somewhat supported by evidence (but the experimental evaluation may be weak, or does not match entirely with the claims, important baselines may be missing, proofs contain important ideas but lack rigor, algorithmic details are only discussed superficially, references are imprecise, assumptions are not sufficiently motivated or explicated, etc.).

**Q2-4 Reproducibility:**

3: Good: key resources (e.g. proofs, code, data) are available and key details (e.g. proofs, experimental setup) are sufficiently well-described for competent researchers to confidently reproduce the main results.

**Q3 Main Strengths:**

- The paper contains interesting and strong results in identifying total effects for two abstractions. It shows both necessary and sufficient conditions for testing the identifiability on ESCGs and SCGs.
- The structure of the paper is clear. The first part of the paper contains the main results, and the second part contains lemmas and examples for the proofs.
- The paper also shows an attempt to minimize the size of adjustment sets, which may be beneficial to practitioners.

**Q4 Main Weakness:**

- In general, adding more intuitions and examples for the key results can be very helpful; in particular, Definition 7, Theorem 3. Perhaps move some lemmas to the appendix.
- Certain definitions are missing or inaccurate, e.g., the definition of backdoor path on SCGs is missing, the definition of summary backdoor criterion uses $Z$ to represent vertices in both ECSG/SCG and their candidate FTCGs (Definition 6)
- The paper can be better organized by adding more explanations (examples) for the main results.

**Q5 Detailed Comments To The Authors:**

Feedback:
1. Definition 5. The problem of *testing* identifiability is different from finding a formula for *computing* the total effect. I would suggest considering a definition that does not mention identifying (do-free) formulas. It may be helpful to check out the definition of identifiability for non-temporal models.

2. Definition 6. Here $Z$ is defined as a set of variables in ESCG/SCG, but it is also used to block the backdoor paths in FTCGs. I don't think the conversion from $Z$ (from ESCG/SCG) to the corresponding variables in FTCG is mentioned in the paper. In fact, the adjustment sets in Theorem 2 and Proposition 3 suggest that this conversion may be non-trivial.

3. I didn't find the definition of backdoor path for SCGs in the paper. This notion of $\sigma$-active backdoor path appears in the main theorem (Theorem 3) and lots of lemmas. It's important to explicitly mention how to check a backdoor path in an SCG -- maybe a formal definition. The paper mentions that $X \leftrightarrow Y$ is the only $\sigma$-active backdoor path of size 2 in SCGs, but isn't $X \leftarrow Y$ also a backdoor path?

4. More intuition is needed for Definition 7 ($\sigma$-blocked path). For example, is it used for characterizing the independence among variables like the d-separation? Examples will be very helpful here.

5. Is there any intuition behind these cases in Theorem 3? Also, more examples will be very helpful.

Question:

6. Is there a reason why Definition 8 (on ambiguous vertices) requires "a descendant of $Y_t$" rather than "a descendant of $X_{t-\lambda}$" as proposed in Definition 6?

Typos:
- Definition 7 second bullet: should "collider at Z" be "collider at W"?
- Proposition 2: The last sentence "In addition, $B_{\gamma}$ is a valid adjustment ..." is repeated.

**Q9 Complying With Reviewing Instructions:**

Yes

---

> ### Author Rebuttal · Authors · 2024-04-05
>
> The reviewer's feedback and comments are deeply valued, and we have addressed all concerns raised, respecting their order of appearance.
>
> Q4 We will provide more examples and illustrations and we will also add a section where we will illustrate the sufficient conditions of identifiability on real world summary causal graphs. These summary causal graphs can be referenced from Aït-Bachir et al., 2023, Peters et al., 2016, and Staplin et al., 2016.
>
> Q5.1 We agree. We will clearly state that we are focusing on the identifiability and not on finding the do-free formula in Theorem 3. Although, in case of identifiability, it is easy to find the do-free formula using our adjustment sets and the Back-Door Adjustment Theorem presented by Pearl, 1993. We will clarify this.
>
> Q5.2 We acknowledge that Definition 6 is unclear. As discussed with the other reviewers, we will simplify Section 3 by removing Theorem 1 and Proposition 1, as they are not necessary for the sufficient conditions of Theorem 3. Therefore, in the final version, we will mainly focus on the sufficient conditions. Please note that the definition of the summary backdoor criterion will not be required to represent all graphs; hence, we will move it to Section 5 and define it specifically for SCGs.
>
> Q5.3 We defined back-door paths for all graphs in page 2. But we agree that it might be unclear so we will give a formal definition of back-door paths in SCGs.
>
> Q5.4 Thank you for pointing out this typo. We meant to say that it is the only sigma-active back-door path of size 2 in the case X is an ancestor of Y (this is the case that we will be interested in within Theorem 3). We will clarify this.
>
> Q5.5 We will give an example for each sufficient condition and we will add a section discussing these conditions on real summary causal graphs.
>
> Q5.6 This is a typo, thank you for pointing it out. We will correct it.
>
>
> References
> * Ait-Bachir, A.; Assaad, C. K.; de Bignicourt, C.; Devijver, E.; Ferreira, S.; Gaussier, E.; Mohanna, H.; and Zan, L. 2023. Case Studies of Causal Discovery from IT Monitoring Time Series. The History and Development of Search Methods for Causal Structure Workshop at the 39th Conference on Uncertainty in Artificial Intelligence.
> * Peters, J.; Janzing, D.; and Scholkopf B. 2013 Causal inference on time series using restricted structural equation models. In Advances in Neural Information Processing Systems 26, pages 154–162.
> * Staplin, N.; Herrington, WG; Judge, PK; Reith, CA; Haynes, R; Landray, MJ; Baigent, C; Emberson, J. 2016. Use of Causal Diagrams to Inform the Design and Interpretation of Observational Studies: An Example from the Study of Heart and Renal Protection. Clin J Am Soc Nephrol.

---

### Official Review · Reviewer_jkxN · 2024-03-19

**Q2-1 Originality-Novelty:** 3
**Q2-2 Correctness-Technical Quality:** 3
**Q2-5 Clarity Of Writing:** 3

**Q1 Summary And Contributions:**

The paper introduces new results on identifiability of total causal effects in two types of time series graphs (ESCGs and SCGs).

**Q2-3 Extent To Which Claims Are Supported By Evidence:**

3: Good: the main claims are supported by convincing evidence (in the form of adequate experimental evaluation, proofs, (pseudo-)code, references, assumptions).

**Q2-4 Reproducibility:**

3: Good: key resources (e.g. proofs, code, data) are available and key details (e.g. proofs, experimental setup) are sufficiently well-described for competent researchers to confidently reproduce the main results.

**Q3 Main Strengths:**

The results in the paper appear to be novel and correct.

**Q4 Main Weakness:**

The paper is very technical and focused on theory, and not so much on the usefulness of the results. I would have liked to see a motivating example of why it is interesting and relevant to consider these types of graphs.

**Q5 Detailed Comments To The Authors:**

(1) Some things were unclear to me:

- What are possible scenarios where we only have access to an ESCG or to an SCG?

- Are these results valid for other types of cyclic graphs? If so, I think this would be important to mention.

- Is Assumption 2 stationarity or something else?

(2) There is no empirical evaluation to support the theory. A small simulation study would have been interesting.

(3) Typos etc:

- In the introduction, the authors refer to ESCGs and SCGs in the second paragraph without having explained the abbreviations (this they do in the third paragraph).

- Definition 6: Should "disjoint" be "distinct"?

- After the definition of $\mathcal{B}_\gamma$ they refer to this as $B_\gamma$

**Q9 Complying With Reviewing Instructions:**

Yes

---

> ### Author Rebuttal · Authors · 2024-04-05
>
> The reviewer's feedback and comments are deeply valued, and we have addressed all concerns raised, respecting their order of appearance.
>
> Q4.1 In order to provide further motivation for our work, we will add a new section focusing on real-world applications. In this section, we will present summary causal graphs from various applications and examine whether the total effect is identifiable based on our Theorem 3. These summary causal graphs can be referenced from Aït-Bachir et al., 2023, Peters et al., 2016, and Staplin et al., 2016.
>
> Q5.1 The first point is answered in Q4.1. We don’t think that our results are valid for any cyclic graph since we used the notion of temporal lag and a specific definition of self-causes in our sufficient conditions. We will clarify this in the text. Yes Assumption 2 is stationarity, we will clarify this.
>
> Q5.2 We don’t think that we can give a convincing simulation study as there are no other methods to compare with (our work is mainly theoretical and in practice it involves knowing if it possible to estimate the total effect). However, we can add a small simulation study based on the summary causal graphs presented in Figure 5.5 which illustrates that the error and variance can be negligible when estimating an identifiable total effect. If the reviewer thinks that this simulation study is interesting, we will add it.
>
> Q5.3 Thank you, we took them into account.
>
> References:
> * Ait-Bachir, A.; Assaad, C. K.; de Bignicourt, C.; Devijver, E.; Ferreira, S.; Gaussier, E.; Mohanna, H.; and Zan, L. 2023. Case Studies of Causal Discovery from IT Monitoring Time Series. The History and Development of Search Methods for Causal Structure Workshop at the 39th Conference on Uncertainty in Artificial Intelligence.
> * Peters, J.; Janzing, D.; and Scholkopf B. 2013 Causal inference on time series using restricted structural equation models. In Advances in Neural Information Processing Systems 26, pages 154–162.
> * Staplin, N.; Herrington, WG; Judge, PK; Reith, CA; Haynes, R; Landray, MJ; Baigent, C; Emberson, J. 2016. Use of Causal Diagrams to Inform the Design and Interpretation of Observational Studies: An Example from the Study of Heart and Renal Protection. Clin J Am Soc Nephrol.

---

### Official Review · Reviewer_83Ki · 2024-03-24

**Q2-1 Originality-Novelty:** 2
**Q2-2 Correctness-Technical Quality:** 3
**Q2-5 Clarity Of Writing:** 2

**Q1 Summary And Contributions:**

The paper under review proposes a necessary and sufficient condition (Theorem 3) for identifying causal effects under causal sufficiency from a time-series summary causal graph and observational distribution. While this presents an interesting contribution to the field, there are several weaknesses that need to be addressed.

**Q2-3 Extent To Which Claims Are Supported By Evidence:**

2: Fair: the main claims are somewhat supported by evidence (but the experimental evaluation may be weak, or does not match entirely with the claims, important baselines may be missing, proofs contain important ideas but lack rigor, algorithmic details are only discussed superficially, references are imprecise, assumptions are not sufficiently motivated or explicated, etc.).

**Q2-4 Reproducibility:**

3: Good: key resources (e.g. proofs, code, data) are available and key details (e.g. proofs, experimental setup) are sufficiently well-described for competent researchers to confidently reproduce the main results.

**Q3 Main Strengths:**

The paper presents interesting results on effect identification in time-series summary graphs, a problem of considerable importance within the realm of causal inference.

**Q4 Main Weakness:**

1) The paper investigates the problem assuming causal sufficiency, which significantly simplifies the involved task but introduces a strong and impractical assumption.

2)  The paper's presentation lacks engagement, as a considerable portion (half of the paper) is devoted to preliminaries and problem setup. Definition 6 could easily be integrated as a brief discussion in the introduction. Additionally, Proposition 1 and Theorem 1 appear to be self-evident statements rather than significant contributions, potentially diminishing their impact.

3) The identifiability result for the extended summary graph (ESCG) in Theorem 2 is straightforward and could be presented as an observation. Note that, given the acyclic nature of ESCGs  and the causal sufficiency assumption, effect identifiability is always achieved by the truncated factorization formula and, consequently, by adjustment over the parents.  The only key observation is that the result must be derived using the densest candidate, and afterward, it suffices to apply Lemma 13 by Jin Tan & Pearl (2003) from their technical report "On the Identification of Causal Effects" (R-290-L 2003).

4) The conditions outlined in Theorem 3 are hard to grasp, although I see similarities with the generalized adjustment criterion for MAGs and PAGs by Perković et al, 2018.  Given the already established characterization of PAGs with cycles (see Bongers et al., 2021), it raises questions regarding the novelty of these results. Nevertheless, it is worth noting that the paper lacks a discussion on this matter and it should at least be addressed as a potential future direction for extending the findings, especially in tackling latent confounding.

* Bongers, S., Forré, P., Peters, J. and Mooij, J.M., 2021. Foundations of structural causal models with cycles and latent variables. The Annals of Statistics, 49(5), pp.2885-2915.

5) As the paper employs the concept of "causal abstraction" as a class of causal models, it overlooks key references on effect identification in Partial Ancestral Graphs (PAGs), such as:

* Perković, E., Textor, J., Kalisch, M. and Maathuis, M.H., 2018. Complete graphical characterization and construction of adjustment sets in Markov equivalence classes of ancestral graphs. Journal of Machine Learning Research, 18(220), pp.1-62.

* Jaber, A., Ribeiro, A., Zhang, J. and Bareinboim, E., 2022. Causal identification under Markov equivalence: calculus, algorithm, and completeness. Advances in Neural Information Processing Systems, 35, pp.3679-3690.

**Q5 Detailed Comments To The Authors:**

See weaknesses and other sections.

**Q9 Complying With Reviewing Instructions:**

Yes

---

> ### Author Rebuttal · Authors · 2024-04-05
>
> The reviewer's feedback and comments are deeply valued, and we have addressed all concerns raised, respecting their order of appearance.
>
> Q.4.1 We agree that the problem of identifying total effects in the presence of cycles and hidden confounding is interesting and we are planning to work on this in future work. We will clarify this in the text. However, we also think that the problem of identifying total effects in the presence of cycles and without hidden confounding is also interesting and provides a first step in the direction of solving the general problem. Also, we argue that in real world applications, there are many studies that consider summary causal graphs without hidden confounding. To illustrate this, we will add a new section where we will illustrate summary causal graphs in many applications and discuss whether the total effect is identifiable according to our theorem 3. Many of these summary causal graphs can be sourced from Aït-Bachir et al., 2023, Peters et al., 2016 and Staplin et al., 2016.
>
> Q.4.2 In response to many reviewers’ feedback, we propose to modify the paper to emphasize its core contribution: sufficient conditions for the total effect to be identifiable in SCGs. Specifically, we intend to simplify Section 3 by removing Assumption 3 and Theorem 1 that are not needed for the sufficient conditions of Theorem 3 ; this will simplify Section 5.1. Furthermore, we plan to interchange Sections 5.1 and 5.2. This entails presenting the sufficient conditions for identifiability in the revised Section 5.1, followed by a discussion of some non-identifiable cases in Section 5.2.
>
> Q.4.3 We agree the result for ESCGs is straightforward; this said, it still important to provide this result as it shows that ESCGs, while providing a summary of the full graph, have the same properties as full graphs when it comes to identifiability in the context retained.
>
> Q.4.4 Bongers et al, 2021 study the properties of SCM with cycles and latent confounding which is not exactly what we are doing in this work. Our work differs from theirs in two main aspects: 1) we focus on a dynamical system and summary causal graphs (which may contain cycles but do not hold the same conceptual meaning as the cycles in an SCM); 2) we investigate the identifiability of the total effect, whereas they explore the identifiability of the SCM (which is more closely linked to direct effects).
>
> Q.4.5 Our study presents a distinct setting compared with Perkovic et al., 2018 and Jaber et al., 2022 for many reasons; for example, in our case, the skeletons of several FTCGs for the same ESCG (or SCG) can vary beyond mere orientation, which is not true for CPDAGs and PAGs. In that sense, our work aligns more closely with Anand et al., 2023, although unlike our approach, Anand et al., 2023 assumes the absence of cycles. We will elaborate on this distinction further in the text. We will delve into all of these related works (mentioned in Q.4.4 and Q.4.5), and we appreciate your mentioning them to us, as we believe that discussing them will help to motivate our own work.
>
> References:
> * Ait-Bachir, A.; Assaad, C. K.; de Bignicourt, C.; Devijver, E.; Ferreira, S.; Gaussier, E.; Mohanna, H.; and Zan, L. 2023. Case Studies of Causal Discovery from IT Monitoring Time Series. The History and Development of Search Methods for Causal Structure Workshop at the 39th Conference on Uncertainty in Artificial Intelligence.
> * Peters, J.; Janzing, D.; and Scholkopf B. 2013 Causal inference on time series using restricted structural equation models. In Advances in Neural Information Processing Systems 26, pages 154–162.
> * Staplin, N.; Herrington, WG; Judge, PK; Reith, CA; Haynes, R; Landray, MJ; Baigent, C; Emberson, J. 2016. Use of Causal Diagrams to Inform the Design and Interpretation of Observational Studies: An Example from the Study of Heart and Renal Protection. Clin J Am Soc Nephrol.
> * Bongers, S., Forré, P., Peters, J. and Mooij, J.M., 2021. Foundations of structural causal models with cycles and latent variables. The Annals of Statistics, 49(5), pp.2885-2915.
> * Perković, E., Textor, J., Kalisch, M. and Maathuis, M.H., 2018. Complete graphical characterization and construction of adjustment sets in Markov equivalence classes of ancestral graphs. Journal of Machine Learning Research, 18(220), pp.1-62.
> * Jaber, A., Ribeiro, A., Zhang, J. and Bareinboim, E., 2022. Causal identification under Markov equivalence: calculus, algorithm, and completeness. Advances in Neural Information Processing Systems, 35, pp.3679-3690.
> * Tara V. Anand, Adele H. Ribeiro, Jin Tian, and Elias Bareinboim. Causal effect identification in cluster dags. Proceedings of the AAAI Conference on Artificial Intelligence, 37(10):12172–12179, Jun. 2023

---

### Official Review · Reviewer_MgHX · 2024-03-28

**Q2-1 Originality-Novelty:** 3
**Q2-2 Correctness-Technical Quality:** 3
**Q2-5 Clarity Of Writing:** 3

**Q10 Ethical Concerns:**

There is no ethical concern.

**Q1 Summary And Contributions:**

This paper studies the identification of total effects between singleton variables, under causal sufficiency, for both extended summary causal graphs and summary causal graphs. This paper has  shown that the total effect is always identifiable for extended summary causal graphs and sometimes identifiable for summary causal graphs under some sufficient conditions. This paper also  provide a set of adjustment sets for estimating the total effect in extended summary causal graphs, and an adjustment set when considering summary causal graphs.

**Q2-3 Extent To Which Claims Are Supported By Evidence:**

3: Good: the main claims are supported by convincing evidence (in the form of adequate experimental evaluation, proofs, (pseudo-)code, references, assumptions).

**Q2-4 Reproducibility:**

3: Good: key resources (e.g. proofs, code, data) are available and key details (e.g. proofs, experimental setup) are sufficiently well-described for competent researchers to confidently reproduce the main results.

**Q3 Main Strengths:**

This  paper studies the conditions of  identification of total effects  under both extended summary causal graphs and summary causal graphs, which makes some contributions to the theory of causal inference. The proposed theories can have wide applications in practice.

**Q4 Main Weakness:**

The paper does not provide the real-world examples and experiments where the proposed method can be used. After identification, this paper does not show how to estimate the total effects.

Assumption 2 requires all the causal relationships remain constant in direction through time, this assumption is very strong and may not be satisfied many practical problems.

**Q5 Detailed Comments To The Authors:**

How to understand that "X causes Y at time t with a time lag of gamma>0 if X=Y'' in Definition 1?

This papers provides  too many graphs and conditions, which are not simple and clear.

Can you show the full name of certain terms when their abbreviations first appear in the paper?

**Q9 Complying With Reviewing Instructions:**

Yes

---

> ### Author Rebuttal · Authors · 2024-04-05
>
> The reviewer's feedback and comments have been received and noted with sincere appreciation. We have made every effort to respond to all concerns raised in Q4 and Q5 in the order they were brought up.
>
> Q4.1 We will introduce a new section dedicated to exploring real-world applications in various domains such as IT monitoring, economy, ecology, and health. In this section, we will illustrate a known summary causal graph for each application and discuss whether the total effect is identifiable according to our theorem 3. Many of these summary causal graphs can be sourced from Aït-Bachir et al., 2023, Peters et al., 2016 and Staplin et al., 2016. We have not included estimations because we couldn't find any known real-world total effects that match our specific setting. Typically, we mainly found real cases where the summary causal graph is known but the total effect is unknown. If the reviewer wants us to provide estimation without comparing to any ground truth, we are willing to provide these estimations.
>
> Q4.2 It is true but if this assumption is not satisfied then the problem of finding a unique total effect would be ill-posed (in a dynamic system with only one multivariate observational time series) since violating the assumption would mean that the total effect would change over time.
>
> Q5.1 It means that Xt-i can cause Xt for all i >0 (but Xt cannot cause Xt); for example the stock price yesterday can affect the stock price today. On the other hand, if X is different from Y then Xt-i can cause Yt for all i >=0 (Xt can cause Yt). We will clarify this in the text.
>
> Q5.2 We will simplify the conditions and focus on our main contribution: providing sufficient conditions for the identifiability of the total effect in SCGs. Specifically, we will streamline Section 3 by removing Assumption 3 and Theorem 1, as they are no longer relevant. This modification will imply the simplification of Section 5.1.
>
> Q5.3 Thank you for your feedback. We have thoroughly reviewed the paper and have made sure to clearly specify the full name of each term when its abbreviation first appears in the paper.
>
> References:
> * Ait-Bachir, A.; Assaad, C. K.; de Bignicourt, C.; Devijver, E.; Ferreira, S.; Gaussier, E.; Mohanna, H.; and Zan, L. 2023. Case Studies of Causal Discovery from IT Monitoring Time Series. The History and Development of Search Methods for Causal Structure Workshop at the 39th Conference on Uncertainty in Artificial Intelligence.
> * Peters, J.; Janzing, D.; and Scholkopf B. 2013 Causal inference on time series using restricted structural equation models. In Advances in Neural Information Processing Systems 26, pages 154–162.
> * Staplin, N.; Herrington, WG; Judge, PK; Reith, CA; Haynes, R; Landray, MJ; Baigent, C; Emberson, J. 2016. Use of Causal Diagrams to Inform the Design and Interpretation of Observational Studies: An Example from the Study of Heart and Renal Protection. Clin J Am Soc Nephrol.

---

### Meta-Review · Area_Chair_f1sp · 2024-04-14

The reviewers were fairly evenly split on this paper: Average Q6 Overall Score: 5.40 (Min: 4, Max: 7) Average Confidence: 3.40 (Min: 3, Max: 4) Number of Forum replies: 12.  The most positive review (by MgHX) was not particularly detailed.  The most thoughtful review (by 83Ki) was negative. I agree with this reviewer that the conditions under which the results are obtained are too restrictive (as also noted by reviewer fkvr).  Therefore, I support a poster presentation only, while encouraging the authors to rewrite the paper by following the many detailed recommendations from the reviewers.